# A handheld photoacoustic microscopic probe integrating a transparent ultrasound transducer and a fiber scanner

Mingyu Ha[1,2,7], Jaewoo Kim [1,2,7], Jihye Lee[3], Seonghee Cho [2,4], Dasom Heo [1,2], Minsu Kim [1,2], Joongho Ahn [2,4,5], Eunwoo Park [1,2], Joo Young Kweon [6], Yuri Kang[1], Yong Joo Ahn [1,2,6], Hyung Ham Kim [1,2,4,6], Won Jong Kim [3] & Chulhong Kim [1,2,4,5,6] ✉

Photoacoustic microscopy (PAM) has been widely used in biomedical studies to provide high-resolution 3D anatomical, functional, and molecular images of living subjects. While handheld PAM systems have been proposed to extend its applicability, it has proved challenging to achieve a compact device that combines fast imaging with high spatial resolution and signal to noise ratio. Here we demonstrate a handheld PAM probe integrating a fiber scanner and high-frequency transparent ultrasound transducer (TUT), called hPAM-TUT. The compact system (measuring 17 mm in diameter, with a 90 mm long rigid body) achieves high lateral and axial resolutions (7 and 47 µm, respectively), has a 2.6 mm diameter field of view, and delivers a single volumetric image in 1.5 s. In living rats, we used hPAM-TUT to visualize various abdominal organs, and in mice we used it to observe epinephrine-induced vascular changes and image the anatomy and functioning of lymphatic vessels after injection of Evans blue dye. Additionally, we successfully delineated murine vascular networks in early metastatic tumors. This handheld PAM probe shows promise for both clinical and research applications in such fields as dermatology, oncology, and intraoperative imaging.

Photoacoustic imaging (PAI) has emerged as a powerful biomedical imaging modality that combines the strengths of optical and ultrasound imaging. By leveraging the photoacoustic (PA) effect, where absorbed light energy is converted into acoustic waves, PAI uniquely provides optical contrast and enables deep tissue imaging beyond the optical diffusion limit. This technique can visualize both endogenous chromophores (hemoglobin, melanin, water, and lipid) and exogenous contrast agents, making it versatile for a wide range of applications, from basic research to clinical diagnostics[1–13]. Among

the various implementations of PAI, photoacoustic microscopy (PAM) stands out for its ability to provide micron-scale volumetric images of living subjects at high speed. PAM excels in mapping vasculatures, measuring blood oxygenation, and detecting specific molecular targets, making it invaluable for studying angiogenesis, tumor microenvironments, and various pathophysiological processes. The non-invasive nature of PAM, coupled with its capability to provide real-time and label-free imaging, has positioned it as a powerful tool in biomedical research, with growing potential for

[1]Department of Convergence IT Engineering, Pohang University of Science and Technology, Pohang, Republic of Korea. [2]Medical Device Innovation Center, Pohang University of Science and Technology, Pohang, Republic of Korea. [3]Department of Chemistry, Pohang University of Science and Technology, Pohang, Republic of Korea. [4]Department of Electrical Engineering, Pohang University of Science and Technology, Pohang, Republic of Korea. [5]Opticho Co. Ltd, Pohang, Republic of Korea. [6]Department of Medical Science and Engineering, School of Convergence Science and Technology, Pohang University of Science and Technology, Pohang, Republic of Korea. [7]These authors contributed equally: Mingyu Ha, Jaewoo Kim. ✉e-mail: chulhong@postech.edu

clinical applications in fields such as dermatology, ophthalmology, pathology, and oncology[14–34].

Despite these advantages, table-top PAM systems face limitations in portability and flexibility that have motivated the development of handheld PAM systems[35–50]. Handheld PAM devices offer several advantages over their benchtop counterparts: (1) increased mobility and ease of use in clinical settings, (2) the ability to image body parts that are difficult to position for viewing by a static system, (3) the potential for point-of-care diagnostics and bedside imaging, and (4) enhanced patient comfort and procedural compliance during imaging. However, developing practical handheld PAM systems has presented three particular challenges: (1) For comfortable handheld operation, the device must be small and lightweight; (2) It must capture images rapidly to minimize motion artifacts and enable real-time visualization; and (3) The handheld form factor must not compromise high imaging performance (signal-to-noise ratio and spatial resolution).

Traditional PAM systems have employed motorized scanning, and consequently their slow point-by-point scanning and bulky motor components hampered the design of handheld devices[51–53]. The introduction of MEMS scanners and galvanometer scanners significantly advanced PAM technology, enabling more compact and faster handheld systems[36,46,47,50]. These systems typically operated by scanning both light and ultrasound simultaneously within a water medium. However, this approach required a trade-off between speed and size: the high drag of water either slowed the scanning[46] or required larger, more powerful scanners[36,47,50] to overcome the water's resistance while maintaining speed. To avoid this compromise, researchers explored optical scanning as a potential solution. Several handheld PAM systems that used optical scanning were introduced, which moved only the laser to obtain images[35,37,39,40,42–45,48,54,55]. While optical scanning avoided the water resistance problem and allowed fast scanning with smaller scanners, it introduced new challenges. Because light and ultrasound could not be coaxially scanned, focused transducers could not be used. The resulting reduction in sensitivity further restricted the use of high-frequency transducers to achieve high axial resolution[35,37,39,40,42–45,48]. Furthermore, optical scanning designs typically employed MEMS or galvanometer mirror scanners, which required rapid displacement of the mirror for scanning. This large moving mass reduced the system's scanning stability and introduced inconsistent image distortions when scanning range or speed is increased. A comprehensive comparison with other handheld photoacoustic probe studies is provided in Supplementary Table S1.

To overcome these limitations and develop a practical handheld PAM probe, here we present an approach combining two key components: a high-frequency transparent transducer (TUT) and a fiber scanner. Fiber scanners, widely used in miniature optical endoscopy research[35,56–63], typically use a two-axis piezoelectric actuator to vibrate a fiber cantilever at the actuator's tip for light scanning. The piezo actuator's tip moves only a few micrometers, while the lightweight fiber cantilever resonates to cover a wide range, enabling fast, stable, and precise scanning in a compact package. Fiber scanners are inherently well-suited for forward-view probe configurations. They achieve wide-range scanning using minimal optical components, which significantly reduces system complexity and simplifies fabrication within the confined probe space. Additionally, their low driving voltage provides enhanced safety for hand-held biomedical applications. TUTs have been developed to seamlessly combine optical and ultrasound imaging in a coaxial manner[16,24,29,64–71]. Previously, due to the opacity of conventional transducers, PAM systems conventionally used opto-ultrasound beam combiners to align the optical and acoustic paths, which inevitably introduced acoustic reflection loss. The increased optical and acoustic paths also limited the optical numerical aperture

and acoustic attenuation respectively. These issues made it even more difficult for optical scanning probes to use high-frequency transducers. Recent work[66] has enabled the development of high-frequency, high-sensitivity, broadband TUTs using transparent matching and backing layers. Using such a high-frequency TUT allows us to coaxially align the optical and acoustic paths without a beam combiner, avoiding the high-frequency loss problems described above.

In this study, we demonstrate a handheld PAM probe that adeptly merges miniaturization, rapid imaging, and high performance. This compact device (referred to as hPAM-TUT) measures just 17 mm in diameter and 90 mm in length. By using a high-frequency TUT and fiber scanner, it achieves an imaging speed of 1.5 s per volume, captures a 2.6 mm diameter field of view. and provides lateral and axial resolutions of 7 and 47 μm, respectively. These design features enable both portability and precision, making the system suitable for a variety of clinical and preclinical applications. The applicability of the system is demonstrated through in vivo imaging of surgically exposed rat abdominal organs, while its rapid speed facilitates the observation of vascular changes induced by epinephrine in a mouse ear and the flow of Evans blue dye within the lymphatic vessels. Additionally, the versatility of this device was highlighted through successful visualization of metastatic tumors in mouse organs, demonstrating its broad applicability for various biomedical imaging needs.

## Results

### Structure and performance of the TUT

Figure 1a shows the structure of the TUT. The piezoelectric material is an LNO crystal, with two transparent front matching layers and a transparent backing layer. Supplementary Text 1 and Supplementary Fig. S1 explain why LNO is the chosen piezo material. Indium tin oxide (ITO) forms the transparent electrode, supplemented by rectangular gold strip electrodes to enhance electrical conductivity. For wire connections, the ground electrode is attached to the brass tube housing using conductive epoxy, while the signal electrode is similarly prepared for a wire connection. Detailed information on layer thicknesses and the TUT fabrication process can be found in the Methods and Materials section. Figure 1b, a photograph of the fabricated TUT, clearly shows various elements of a film target—large and small text, symbols, and grid patterns—that demonstrate the TUT's ability to provide clear optical transmission, which is crucial for optical scanning. Figure 1c depicts the light transmittance of the TUT across a range of wavelengths. The TUT exhibits high transmittance in both the visible and infrared regions, with a measured transmittance of 64.3% at the 532 nm wavelength used in this work.

Figure 1d shows the simulated pulse-echo response of the TUT based on its design. The center frequency is 33.5 MHz, with a −6 dB fractional bandwidth of 68.3%. Figure 1e presents the measured acoustic properties of the fabricated TUT. The center frequency is 37.2 MHz, with a −6 dB fractional bandwidth of 52.2%. The simulated acoustic intensity field shows the TUT's even sensitivity over a 4 mm × 4 mm area on an imaging plane. Yet, measured acoustic field showed that TUT could receive signal evenly in about 2.6 mm range (Supplementary Fig. S2). The TUT's transparency, its wide bandwidth in the high-frequency range, and its sensitivity all make it well-suited for high-performance PAM imaging using optical scanning techniques.

### Fiber scanner and optical scanning protocol

Figure 2a shows a schematic and photograph of the fiber scanner. The actuator is a piezoelectric tube with ±XY electrodes; a single-mode fiber cantilever is attached to its end. When a periodic sinusoidal wave, resonant with the natural frequency of the fiber cantilever, is applied to the ±XY electrodes, the oscillation of the fiber cantilever is amplified, allowing for optical scanning over a larger range than the mechanical movement of the actuator. The scanning frequency is determined by

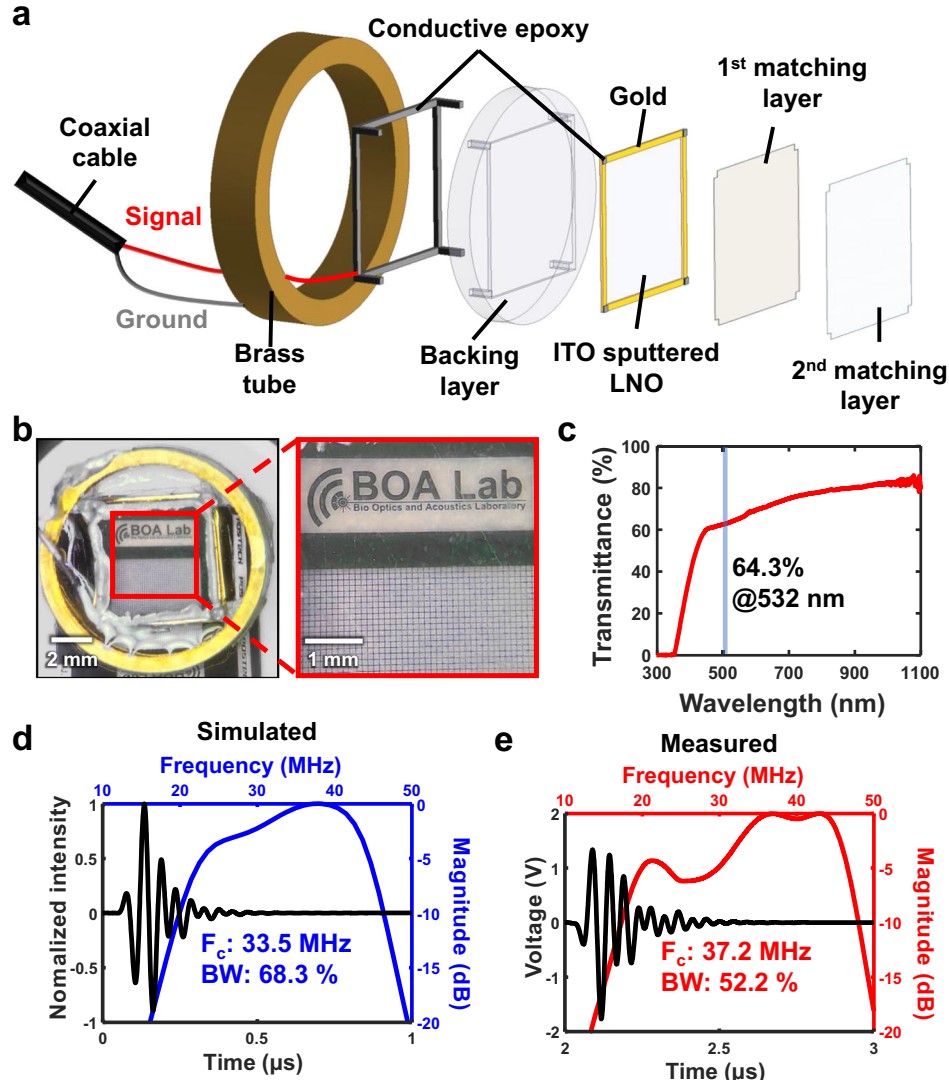

**Fig. 1 | Characteristics of a high-frequency transparent ultrasound transducer (TUT). a** Exploded view of the TUT. **b** Photograph of the TUT lying atop an imaging test target. **c** Measured optical transmittance. **d** Simulated and **e** measured acoustic properties. ITO indium tin oxide, LNO lithium niobite, Fc center frequency, BW bandwidth.

the resonant frequency of the fiber cantilever, expressed by

$$\omega = \beta \left(\frac{E}{\rho}\right)^{\frac{1}{2}} \frac{R}{L^2} \tag{1}$$

where $\beta$ is a mode-related constant, and $E$, $\rho$, $R$, and $L$ represent Young's modulus, the mass density, the radius, and the length of the fiber cantilever, respectively[57]. As the length of the fiber cantilever decreases, the resonant frequency increases, allowing for faster scanning. However, shortening the cantilever length can decrease the scanning stability and lead to uneven scanning coverage due to aliasing, depending on the laser pulse repetition frequency (PRF). To optimize the scanning frequency, we conducted simulations assuming the use of a 500 kHz laser system (Supplementary Fig. S3). This simulation has been experimentally shown in Supplementary Fig. S4. The fill factor within the scanning area increases as the frequency rises, but aliasing occurs above 300 Hz and the fill factor fluctuates above 400 Hz. Based on these simulation findings, to set the resonant frequency in the 300–350 Hz range, we set the length of the fiber cantilever to 18 mm. Resonance was noted between 327 and 335 Hz in this range, and we chose 333 Hz as the scanning frequency because it

resulted in the most orthogonal scanning along the x and y axes, with minimal crosstalk between the two directions.

Figure 2b illustrates how the fiber scanner is driven in a spiral scanning pattern. By applying two voltages with a 90° phase difference to the ±XY electrodes at the resonant frequency of the fiber cantilever, and then gradually increasing the voltage amplitude over time, we generated a spiral pattern, causing the fiber scanner to oscillate. Imaging scan is conducted over 1 s (blue in Fig. 2b), followed by 0.2 s of active breaking (red in Fig. 2b), where the fiber is decelerated by applying a 180° phase-opposed voltage, and 0.3 s of free decay (green in Fig. 2b), allowing the fiber to naturally dampen, resulting in a full scan every 1.5 s[56]. We simulated the effect of the laser PRF on the light density during spiral scanning within the experimental field-of-view (FOV) (Fig. 2c). To achieve fast scanning over a wide area while maintaining high resolution, a laser with a high PRF is required. However, spiral scanning concentrates light in the central region during the initial scanning phase, significantly increasing the energy density and potentially damaging the target. For example, with a fixed PRF of 500 K, the laser energy is highly concentrated in the center. A lower fixed PRF results in a slower imaging speed (Supplementary Fig. S5). To solve this issue, the PRF was customized to increase in steps over time,

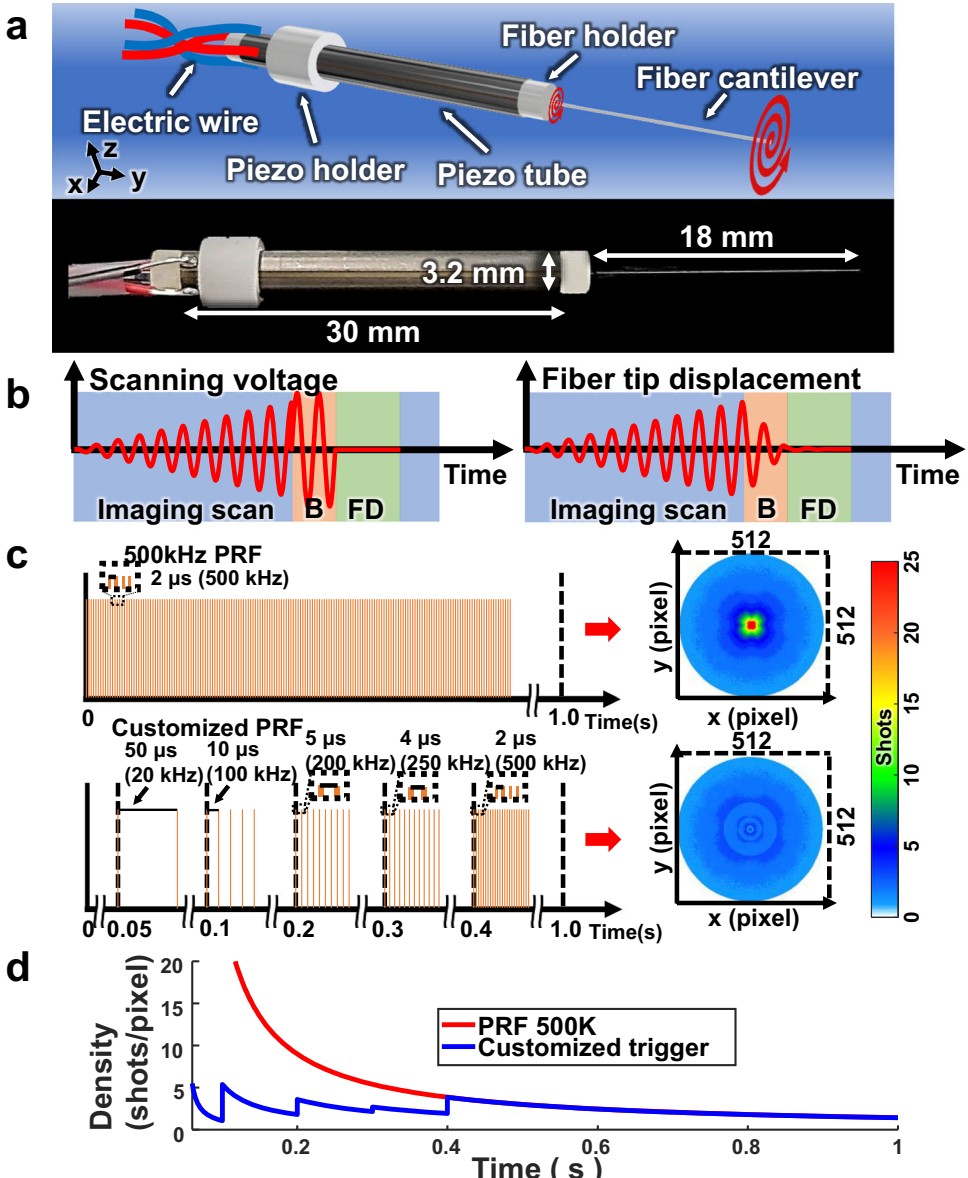

**Fig. 2 | Schematic and scanning protocol of the fiber scanner. a** Schematic and photograph of the fiber scanner. **b** Scanning pattern and resulting tip displacement of the fiber scanner. **c** Comparison of laser trigger rate and simulated laser densities with a fixed PRF of 500 kHz and a range of customized PRF settings. Left, PRF setting; right, areal illumination density. **d** Graph of density (shots/pixel) over time corresponding to the settings in (**c**). B active breaking, FD free decay, PRF pulse repetition frequency.

such as 20 KHz for 0.05–0.1 s, 100 KHz for 0.1–0.2 s, 200 KHz for 0.2–0.3 s, 250 KHz for 0.3–0.4 s, and 500 KHz for 0.4–1 s. In this way, the light density within the FOV was more evenly spread out, minimizing or eliminating potential sample damage. The graph of shots per pixel over time numerically demonstrates the improved uniformity in energy concentration. Comparative imaging results between the fixed PRF mode and the variable PRF mode are presented in Supplementary Fig. S6. Details on laser safety are provided in Supplementary Text 2.

### Description of the hPAM-TUT probe and the system's performance

Figure 3a depicts the system schematically. A 532 nm laser beam enters a collimator and is transmitted into one end of a single-mode fiber. The light emerging from the fiber cantilever at the other end passes through three lenses, two for collimating and one for focusing. The light then passes through the TUT, the water tank, and a film, finally forming an optical focus at 0.25 mm from the film. Each component is

fixed with housing. The probe has an outer diameter of 17 mm, with a rigid section measuring 90 mm in length. The remaining portion is flexible, consisting of wires and optical fiber jacket. The entire probe weighed 11 g, which was sufficiently small and light to allow handheld imaging.

A data acquisition board (DAQ) controls the hPAM-TUT system, following the timing diagram in Supplementary Fig. S7. Scanning signals are generated by the DAQ and subsequently amplified by a piezo amplifier. These signals are then delivered to the fiber scanner, where they create a spiral scanning pattern. The laser pulsing and data acquisition are also controlled and synchronized by the DAQ, which generates customized triggers with a designated delay. Through this synchronized control, the PA signals are captured by the TUT, amplified, and then delivered to a digitizer for volume data acquisition.

To demonstrate the performance of the hPAM-TUT probe, imaging tests were conducted using a film phantom. Images were

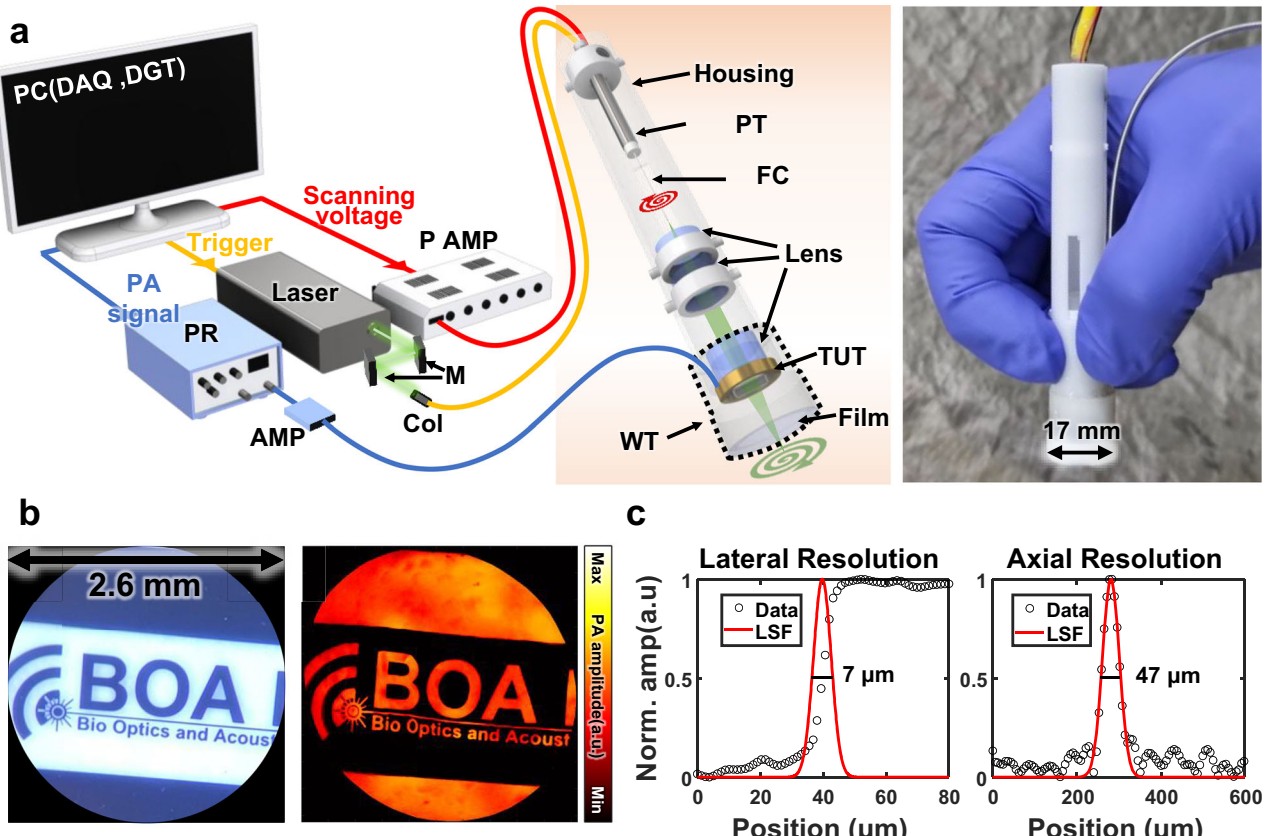

**Fig. 3 | Schematic and performance of a handheld PAM probe with a TUT and fiber scanner (hPAM-TUT). a** Schematic and photograph of the hPAM-TUT. **b** Target photograph & PA image. **c** PA lateral and axial resolutions. DAQ data acquisition device, DGT digitizer, PR pulser/receiver, P AMP piezo amplifier, AMP amplifier, M mirror, Col collimator, PT piezo tube, FC fiber cantilever, WT water tank, ESF edge spread function, LSF line spread function.

reconstructed based on position sensor data (details in Methods section and Supplementary Fig. S8). As shown in Fig. 3b, the PA image closely matches the original photograph. The imaging FOV was determined to be a circle with a diameter of 2.6 mm. The lateral and axial resolutions−7 and 47 μm, respectively−were measured by extracting and fitting the line spread functions from the image. One volumetric PA image was acquired in 1.5 s, and the signal to noise ratio (SNR) was 38 dB. Depth-dependent resolution and SNR curves measured in water and tissue-mimicking phantoms are presented in Supplementary Text 3 and Supplementary Fig. S9. In addition, we evaluated the uniformity of the optical excitation and ultrasonic receiving regions within the FOV by imaging black tape to generate a signal intensity map. The difference between the strongest and weakest signals in the image was within 1.85 dB (Supplementary Fig. S10). The scanning accuracy of the piezoelectric tube was maintained even during long-term operation exceeding 1 h (Supplementary Fig. S11).

### In vivo PA imaging of a rat's abdominal organs

To validate the performance and applicability of the hPAM-TUT, we obtained PA images of surgically exposed abdominal organs in rats, specifically the intestine, stomach, cecum, and bladder. The handheld imaging process is shown in Supplementary Movie S1. Figure 4a presents a photograph of each organ, captured immediately after image acquisition. Figure 4b shows the PA maximum amplitude projection (MAP) images processed from the reconstructed volume data, which correlate well with the photographs in Fig. 4a. Across all organs, we observed arteries and veins running parallel to each other, as well as a distinct capillary network. In the depth-encoded image shown in Fig. 4c, thin capillaries are observed near the surface of the organs, while thicker vessels are found deeper within the muscular layers. Figure 4d shows rendered volume images, where the overlapping vessels are clearly distinguishable due to the excellent axial resolution (Supplementary Movie S2).

In Fig. 4b1, c1, showing the intestine, we identify arterial vessels that likely represent the jejunal artery branches, along with venous branches[72,73]. The blood vessels in the intestine exhibit a highly tortuous, zigzag pattern, with thin and densely packed capillaries near the surface. This specialized vascular structure is essential to support efficient nutrient absorption and supply, highlighting the adaptability of the intestinal vasculature to meet the demands of the digestive processes[72,74]. In Fig. 4b2, showing the stomach, we observe vessels likely branching from the gastroepiploic artery. Figure 4c2, a depth-encoded image, clearly shows the vessels of the serosa running over parallel arteries and veins. In Fig. 4b3, which shows the cecum, we identify vessels branching from the anterior cecal artery[72,73,75] and loop structured vasculature. This loop formation is also evident in Fig. 4d3, a 3D rendered image. In Fig. 4b4, which shows the bladder, we observe branches of the vesical artery[76].

These results highlight the utility of hPAM-TUT. The system's rapid imaging speed and compact design enable minimally invasive handheld imaging, which was previously challenging to achieve. Furthermore, the system's high-resolution imaging provides excellent visualization of vascular 3D anatomy. These characteristics demonstrate hPAM-TUT's potential as a tool for intraoperative guidance and surgical decision-making. The combination of minimal invasiveness and superior imaging performance positions hPAM-TUT as a valuable advance in surgical imaging technology.

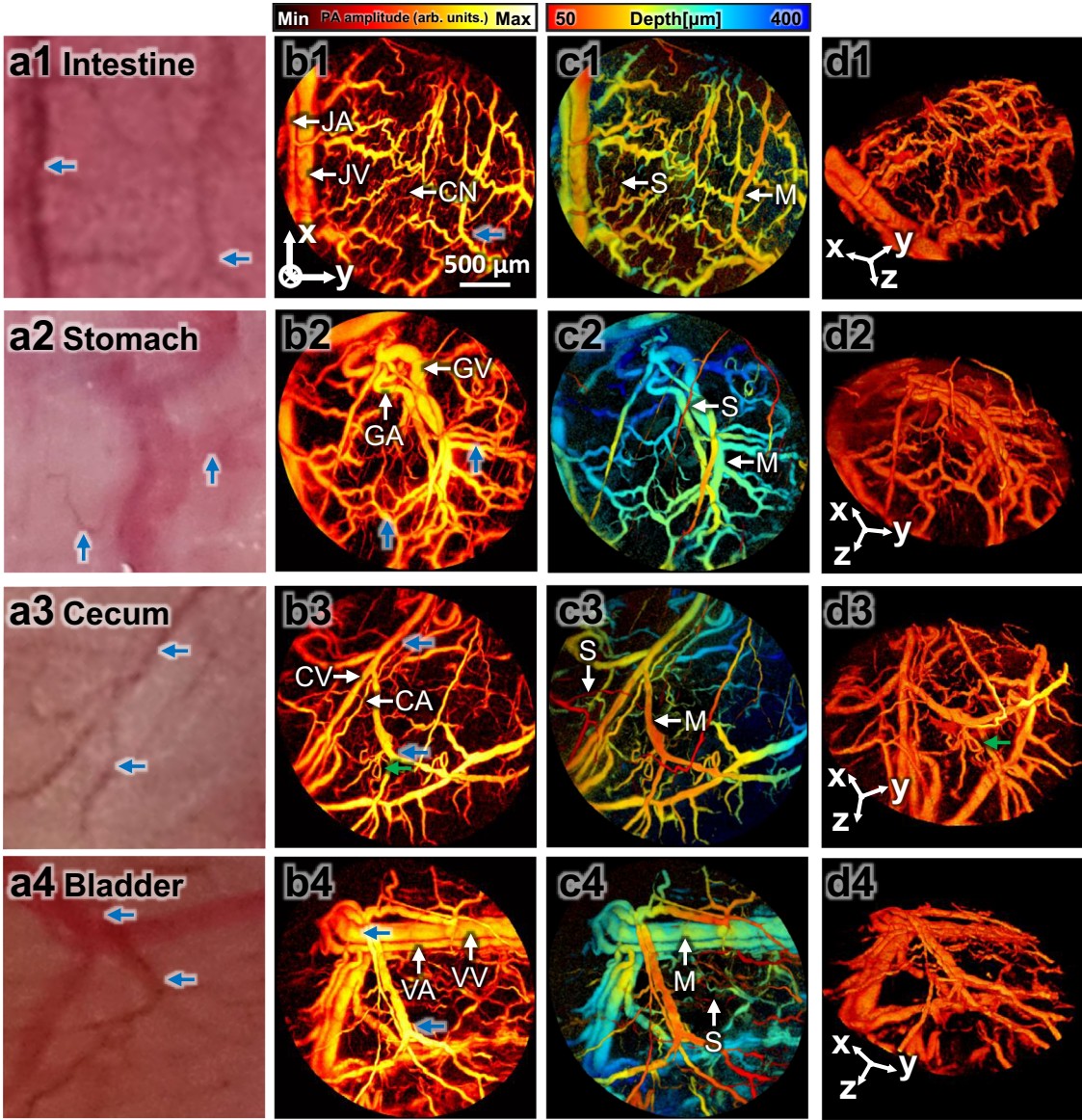

**Fig. 4 | PA images of surgically exposed internal organs in a rat in vivo.** The numbers in the alphanumeric figure labels are code for the following organs: 1, intestine; 2, stomach; 3, cecum; and 4, bladder. The letters coded as **a** Photographs of vasculature in the organs. **b** PA maximum amplitude projection (MAP) images of microvasculatures. **c** Depth-encoded PA images. **d** 3D PA images (Supplementary Movie S2). Blue arrows indicate corresponding structures in photographs and PA images, and green arrows indicate looped vasculatures. JA jejunal artery, JV jejunal vein, GA gastroepiploic artery, GV gastroepiploic vein, CA cecal artery, CV cecal vein, VA vesical artery, VV vesical vein, CN capillary network, S serosa vessel, and M muscularis vessel.

## In vivo monitoring of epinephrine-induced vasoconstriction in a mouse ear

To demonstrate fast imaging with the hPAM-TUT and show its potential for quantitative analysis, we monitored the vascular changes in a mouse ear following epinephrine injection. This experiment was repeated in five mice to ensure reproducibility. Figure 5a shows a series of PA MAP images from the entire FOV, before and after the injection. Both vessel diameter and PA amplitude decrease notably, particularly in microvessels (blue arrows, under 100 µm) (Supplementary Movie S3). Figure 5b quantifies the changes in vessel density and PA amplitude, revealing a rapid reduction in both parameters immediately after the injection. Details on the quantification method can be found in Supplementary Text 4 and Supplementary Fig. S12. The vessel density decreases by ~60% overall, with the microvessels showing up to 68% constriction and the macrovessels exhibiting a maximum reduction of about 26%. These findings demonstrate the strong vasoconstrictive response induced by epinephrine, mediated by alpha-adrenergic receptors in vascular smooth muscle. Arterioles, being thinner and containing more smooth muscle than venules, contract more significantly, as noted in earlier reports[77–79]. Additionally, the PA signals decrease by up to 6.5 dB in the microvessels but only 3.3 dB in the macrovessels. As the vessels constrict, blood flow diminishes, reducing the overall hemoglobin concentration. This observation emphasizes hPAM-TUT can not only resolve structural changes but also infer dynamic hemodynamic alterations.

Capillaries show an even faster response, with some vessels losing their signals immediately after the injection due to reduced blood flow, indicating their heightened sensitivity to flow changes. As shown in Fig. 5c, the vascular signal starts diminishing within 1.5 s post-injection and disappears entirely after 4.5 s. Figure 5d presents a quantitative graph of the vascular area over time, showing a steep decline within the first few seconds, providing a detailed temporal profile of the vasoconstriction process. This analysis is crucial for studying the kinetics of vascular responses to drugs like epinephrine.

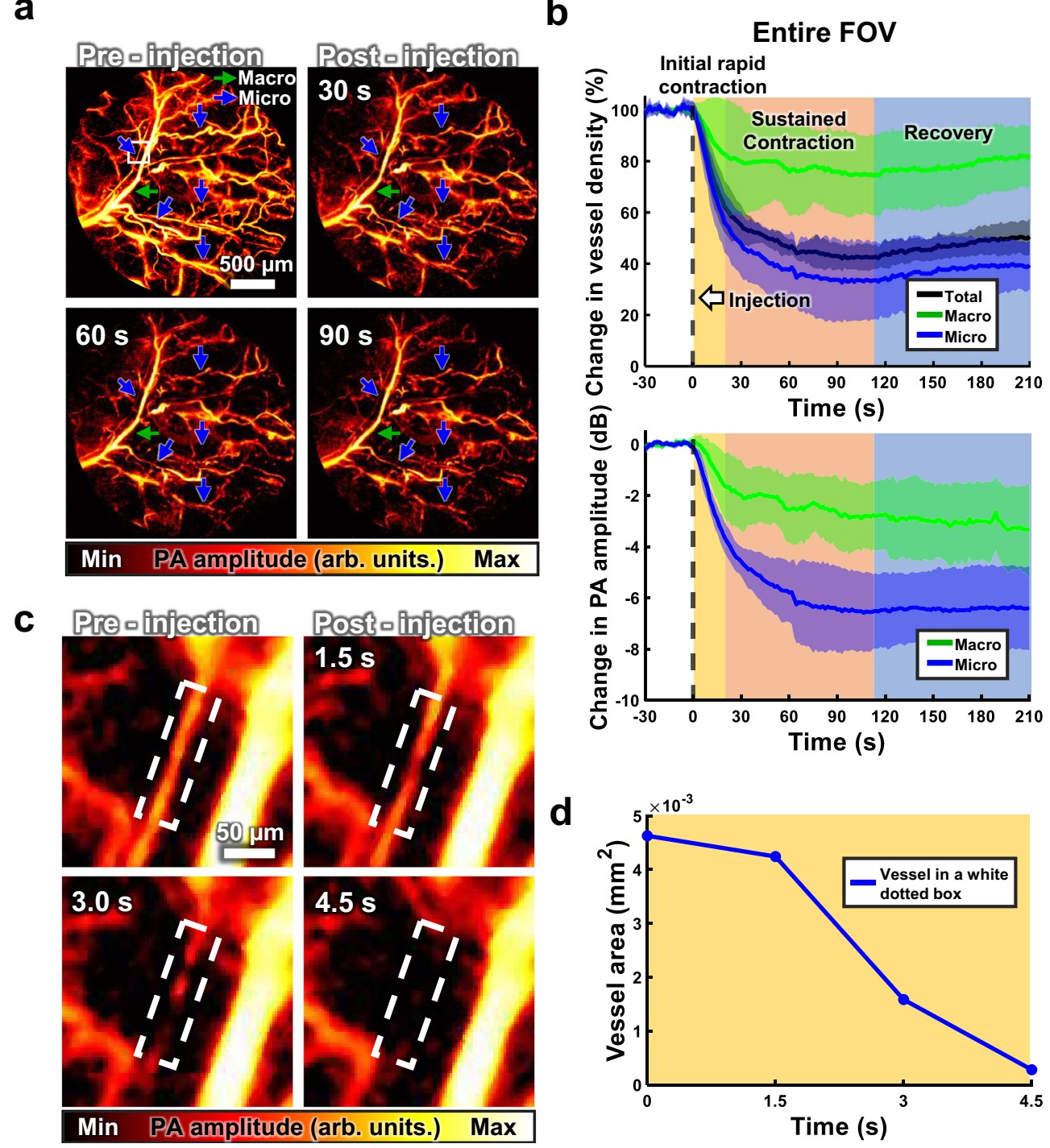

**Fig. 5 | Epinephrine-induced vasoconstriction in the vasculature of a mouse's ear. a** PA MAP images of the vasculature at pre-injection and 30-, 60-, and 90-seconds post-epinephrine injection (Supplementary Movie S3). **b** Quantitative analysis of changes in vessel density and PA amplitude of total, macro (>100 μm), and micro (<100 μm) vessels across the entire region. Data are presented as mean ± 95% CI (*n* = 5). **c** Magnified images of the white box in (**a**). **d** Quantitative analysis of change in the vascular area in the white dashed box in (**c**). CI confidence interval.

This finding emphasizes the advantage of hPAM-TUT's high temporal resolution, enabling the capture of such transient physiological phenomena. The vessel contraction occurs within seconds, a speed at which conventional imaging modalities would likely miss the majority of these events. The ability to track these changes in real time demonstrates the effectiveness of our imaging approach in studying dynamic vascular responses.

## In vivo PA monitoring of lymphatic vessels after injection of Evans blue in a mouse ear

To demonstrate the potential applications of contrast-enhanced imaging, we monitored lymphatic vessels in mouse ears following Evans blue dye injection (Fig. 6). Subcutaneously injected Evans blue dye binds to albumin in the interstitial plasma[80]. The interstitial albumin travels through lymphatic vessels, entering initial lymphatics through

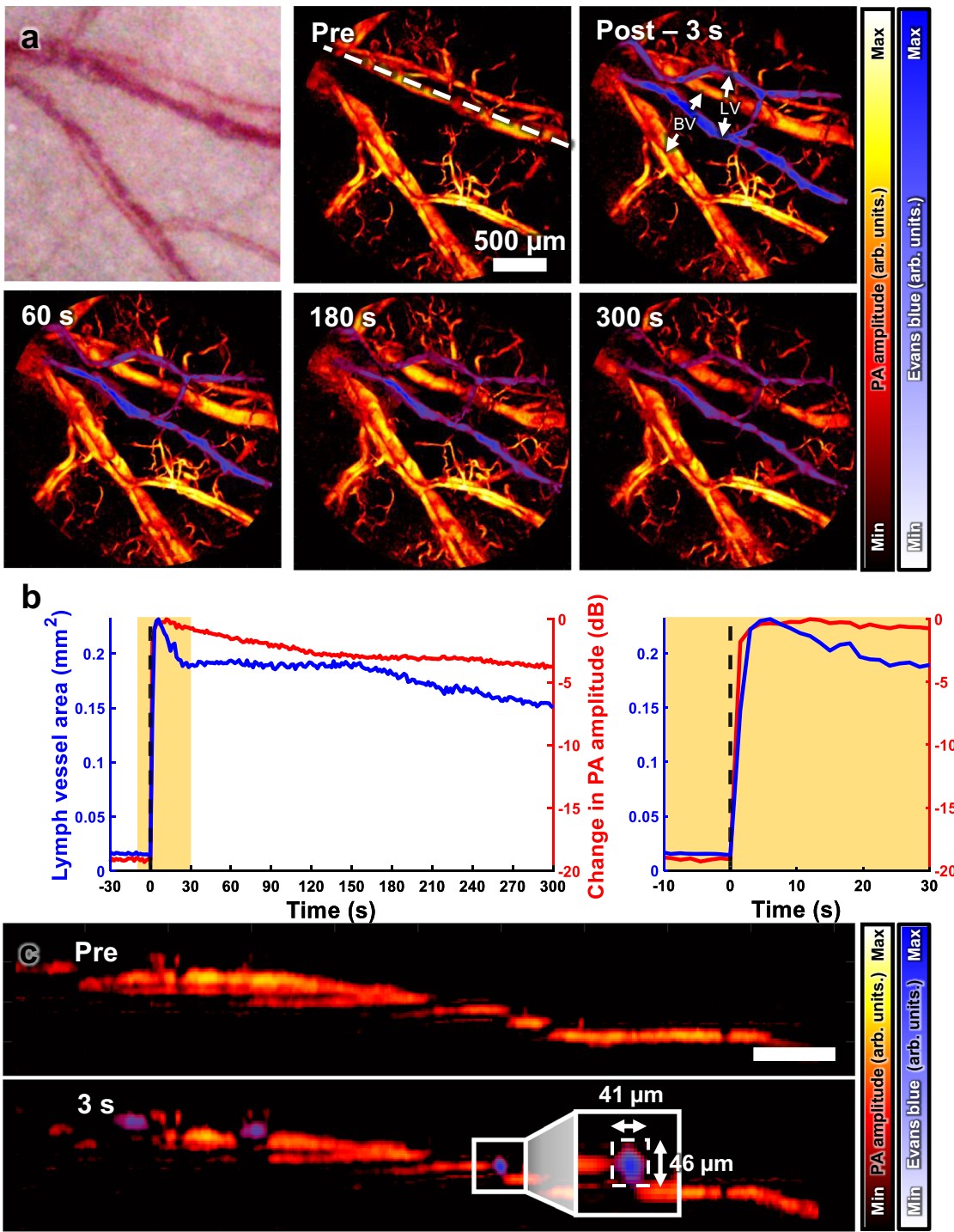

**Fig. 6 | PA imaging of lymphatic vessels in a mouse ear with Evans blue injection. a** Photograph and PA MAP images of blood and lymphatic vessels in the ear at pre-injection, 3-, 60-, 180-, and 300-seconds post-injection of Evans blue (Supplementary Movie S4). **b** Quantitative analysis of the change in lymph vessel area and its PA amplitude. Left, graph for the entire time period, and right, zoomed-in graph of the time range highlighted in the left graph. **c** Cross-sectional B-scan images along the white dotted line in (**a**). BV blood vessel, LV lymphatic vessel.

unidirectional valve-like structures when interstitial pressure is elevated[81]. As shown in Fig. 6a, the vascular networks are well correlated in the photograph and the pre-injection PA MAP image. Following the Evans blue injection, previously unobservable lymphatic vessels rapidly appear. This rapid appearance can be attributed to increased interstitial pressure from volume expansion, driving dye entry into the lymphatic vessels. The newly visible structures exhibit the distinct interconnected mesh-like pattern of lymphatic networks. Figure 6a shows progressive decreases in both lymphatic vessel area

and PA signal intensity over time (Supplementary Movie S4). This decline is likely caused by gradual Evans blue clearance through the lymphatics and decreasing interstitial pressure.

Figure 6b quantifies the changes in lymphatic vessel area and PA amplitudes. Details on the quantification method can be found in Supplementary Text 5 and Supplementary Fig. S13. The lymphatic vessel area peaks within 6 s post-injection, rapidly decreases to 80% by the 30-s timepoint, then gradually declines to 65%. This pattern suggests initial expansion due to injection pressure, followed by

stabilization. The PA amplitude of the lymphatic vessel peaks at 12 s, then gradual decreases, and finally shows a −4 dB signal reduction to peak value. There was a 6 s delay between the peak of the PA amplitude and the peak of the lymphatic area. This delay is likely due to the initial mixing of the existing plasma and dye in the lymph vessel, decreasing the dye concentration and delaying the rise of the PA amplitude.

To identify the relative depth positions of blood vessels and lymph vessels in the overlapped area, we analyzed the cross-sectional B-scan images. Figure 6c shows pre- and post-injection B-scan images extracted along the white dotted line in Fig. 6a. Post-injection images reveal emerged lymphatic vessels. Comparing the positions of the blood and lymphatic vessels demonstrates that the lymphatic vessels are located immediately above the blood vessels. Despite these vessels' close proximity due to the thin ear structure (~250 μm), the system's high lateral and axial resolution enable a clear distinction between them, as shown in the magnified view in Fig. 6c.

These imaging results highlight several key aspects of contrast agent application in the hPAM-TUT the system. Simultaneous imaging of blood and lymphatic vessels provides comprehensive information about tissue microcirculation and lymphatic drainage. The distinct responses of vascular and lymphatic systems after injection demonstrate their different roles and functions. The temporal observation of dye signal reduction demonstrates the system's real-time sensitivity to dye concentration changes.

**In vivo PA imaging of metastatic tumors in a mouse's abdomen**
To demonstrate the applicability of the hPAM-TUT system and its potential for early cancer diagnosis via laparoscopy, we performed in vivo imaging on a mouse with metastatic cancer. Fourteen days after intrasplenic injection, metastatic tumors had developed in the mouse, and imaging was conducted using the hPAM-TUT system. Figure 7a shows a photograph of the cancerous pancreas region, while Fig. 7b presents the PA MAP image of the same area, demonstrating high

correlation with Fig. 7a. Notably, small microvasculatures, which are invisible in the standard photograph, are clearly visualized in the PA images (blue arrows in Fig. 7b). Vascular dysplasia and high tortuosity, commonly observed near cancerous regions, are also well identified[82]. The presence of tumor in the corresponding area was verified through H&E stained histology images. (Fig. 7c).

To compare the vascular structure between the cancerous and normal areas, the experiment was repeated with a normal mouse under the same experimental conditions. Figure 7d shows an optical image of the pancreas in the normal mouse, and Fig. 7e displays the corresponding PA MAP image. Comparing Fig. 7b and e, the cancerous region exhibits higher vascular density, thinner vessels, and more complex structures. Specifically, highly tortuous vessels are clearly observed in the green boxed area magnified in Fig. 7b. For quantitative comparison, four vascular parameters were analyzed: the vascular area density (VAD), vascular skeleton density (VSD), vascular diameter index (VDI), and vascular complexity index (VCI) (Fig. 7f). For detailed explanations of each parameter and analysis method, see Supplementary Text 6 and Supplementary Fig. S14[83]. The VAD in the cancerous region is $0.30 \pm 0.03$, nearly 2.1 times higher than the value of $0.14 \pm 0.02$ observed in the normal region, indicating a significantly larger total vascular area. The VSD, representing the total vascular length, is $0.028 \pm 0.004$ in the cancerous region, 2.5 times higher than the value of $0.011 \pm 0.0001$ in the normal region. In contrast, the VDI, representing mean vessel diameter, is $27.9 \pm 3.0$ μm in the cancerous region, ~85% of that ($33 \pm 4.1$ μm) observed in the normal region. The VCI, reflecting vascular structural complexity, is $306 \pm 103$ in the cancerous region, 2.6 times higher than that ($118 \pm 7$) observed in the normal region. The increase in VSD surpasses that of VAD, while VDI shows a decrease. This indicates that the expansion in vascular area is primarily due to the new formation of small blood vessels. The dramatic increase in VCI further indicates the formation of highly complex neovascularization.

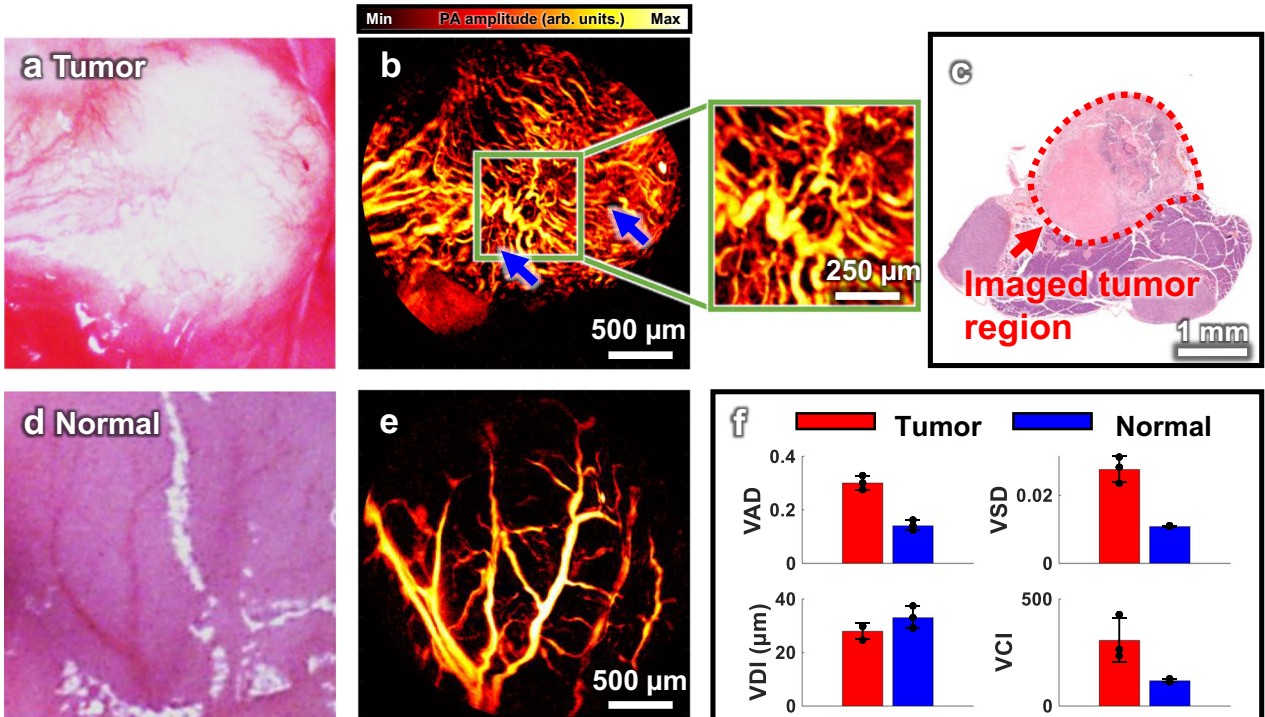

**Fig. 7 | PA imaging of a metastatic tumor in a mouse's abdomen. a** Photograph of metastatic tumor region in the pancreas. **b** PA MAP image and magnified view of microvasculature of the tumor region in (**a**). **c** Photograph of excised H&E stained slide from the tumor region. **d** Photograph of normal region in the pancreas. **e** PA MAP image of microvasculature of the pancreas at (**d**). **f** Comparison of vessel metrics. Data are presented as the mean ± standard deviation, $n = 3$. VAD vessel area density, VSD vessel skeleton density, VDI vessel diameter index, VCI vessel complexity index.

These imaging results highlight the diagnostic potential of the hPAM-TUT system. Unlike traditional PAM systems, hPAM-TUT provides efficient, minimally invasive in vivo imaging of surgically exposed internal organs in animals. The system successfully acquired high-resolution images of early-stage metastatic tumors, showcasing its potential for early cancer diagnosis in laparoscopic settings. In addition, the analysis of various vascular indices indicates hPAM-TUT's significant potential for investigating and diagnosing vascular-related diseases.

## Discussion

The development of handheld PAM represents a significant advancement in expanding the applicability of PAM in preclinical and clinical settings. A practical handheld PAM system must simultaneously address three critical factors: size, imaging speed, and image quality. Our study presents a approach that successfully balances these factors, overcoming the limitations of previous designs. Early handheld PAM systems employed scanners in water to maintain coaxial alignment of light and ultrasound, achieving high image quality but at the cost of compromised size and imaging speed. The introduction of optical scanning methods resolved the issue of water resistance and improved both size and imaging speed. However, this approach made it unfeasible to use focused transducers, reducing sensitivity and limiting the use of high-frequency transducers. Additionally, a beam combiner, necessitated by opaque transducers, extended the optical and acoustic paths, resulting in constraints on optical numerical aperture and high-frequency signal loss, compromising resolution. Moreover, the MEMS and galvanometer scanners commonly employed in these systems presented stability issues, further degrading imaging speed and quality.

Our design addressed these challenges by integrating a high-frequency TUT with a fiber scanner. The system achieves impressive specifications: a compact size (17 mm diameter, 90 mm rigid body length), rapid imaging speed (1.5 s per volume), and high image quality (7 μm lateral resolution, 47 μm axial resolution). The TUT eliminates the need for a beam combiner for aligning optical and acoustic paths, facilitating high-frequency signal acquisition and enhanced axial resolution. The fiber scanner, utilizing a resonant fiber cantilever, ensures stable, high-speed scanning over a wide area within a thin probe.

We demonstrated the system's versatility across multiple applications. First, we imaged the external surfaces of rat abdominal organs in a minimally invasive manner, a challenging task for conventional PAM. Its high lateral resolution provided clear visualization of capillaries, while its high axial resolution captured depth information and differentiated closely adjacent blood vessels. The resulting volume-rendered images revealed intricate vascular networks, demonstrating the system's capability to capture complex 3D vascular structures previously unattainable with handheld PAM systems. Second, we observed epinephrine-induced vasoconstriction in mouse ears, highlighting the system's capacity for rapid hemodynamics monitoring. The images clearly displayed reductions in blood vessel diameters, and vessel area analysis across the imaging field revealed the distinct phases of contraction and recovery induced by epinephrine injection. Magnified views highlighted the dramatic constriction of capillaries, with some vessels nearly disappearing within just four frames (6 s). The high lateral resolution (7 μm) and rapid image acquisition speed (1.5 s per volume) of our system enabled this level of detailed observation, demonstrating the potential of hPAM-TUT for studying vascular responses to pharmacological interventions. Building on its success in vascular imaging, we explored the system's potential in lymphatic imaging using Evans blue dye. Following subcutaneous dye injection, previously undetected lymphatic vessels were rapidly visualized, showcasing the system's ability to capture dynamic physiological activities. Quantitative analysis revealed the initial expansion of

lymphatic vessels, followed by a gradual reduction in lymph area and signal intensity. The system's superior resolution distinguished lymphatic vessels from nearby blood vessels, underscoring its capability for detailed imaging of both blood and lymphatic systems. These results emphasize the versatility of hPAM-TUT in studying the microcirculation and lymphatic drainage, offering valuable insights into tissue physiology and expanding its applications beyond vascular imaging. Finally, we applied the hPAM-TUT system to visualize vascular structures in a metastatic tumor model. Using a compact handheld device, we successfully imaged early metastatic tumor regions, which are difficult to observe with conventional PAM systems, and compared them with vascular structures in normal tissues. Tumor regions exhibited increased vascular density characterized by numerous thin, tortuous vessels. Quantitative analysis using vascular metrics (VAD, VSD, VDI, and VCI) validated these observations, highlighting the system's potential to diagnose tumor in the early stage. Moreover, the detailed analysis of vascular metrics demonstrates its capability for diagnosing and assessing various vascular-related diseases.

Despite these achievements, opportunities for further advancements remain. First, additional miniaturization is feasible. Previous studies have developed a fiber scanner endoscope with a diameter of ~1 mm[56] and a TUT measuring 1 mm × 1 mm[65]. Combining these advances could enable the creation of a front-view endoscope with a diameter under 2.5 mm, compatible with conventional endoscopy systems. Second, the high transmittance of our TUT across the visible and NIR wavelengths opens avenues for multi-wavelength photoacoustic imaging, enabling functional studies such as oxygen saturation measurements. Additionally, integrating contrast agents, photothermal therapy agents, and disease- or cell-targeting molecules could broaden diagnostic and therapeutic applications beyond blood vessels, nerves, and lymph nodes. Third, the integration of other optical modalities holds significant potential. Fiber scanners have been applied in imaging techniques such as white light endoscopy (WLE), optical coherence tomography (OCT), and near-infrared spectroscopy (NIRS). TUTs have also facilitated multimodal research. The structure of the hPAM-TUT system supports seamless incorporation of additional modalities by accommodating various fibers and detectors. This adaptability could lead to the development of a multi-functional imaging probe suitable for diverse clinical scenarios, offering unprecedented flexibility in medical imaging. Fourth, current dynamic experiments are performed by mounting the probe on a fixed stage to precisely quantify microvascular contraction. However, motion artifacts caused by hand tremor or patient movement are unavoidable in future clinical applications. To minimize these issues, from a hardware perspective, increasing the image frame rate is crucial for acquiring less distorted images. This can be achieved by increasing the resonant frequency through increasing the mechanical stiffness of the optical fiber, for example, by enlarging the cladding diameter or applying a metal coating, thereby enabling faster scanning. Such hardware-based approaches are expected to effectively suppress motion artifacts occurring in clinical settings and provide stable imaging performance.

In summary, the hPAM-TUT PAM system, combining a fiber scanner and high-frequency TUT, successfully addresses the challenges of size, imaging speed, and image quality. In particular, fiber resonant scanning offers high-speed, electromagnetically safe, and long-term stable performance, making it more clinically viable than MEMS-based scanners. We have demonstrated its utility in 3D imaging of internal organs, monitoring of drug-induced vascular changes, lymphatic imaging with contrast agent, and analyzing metastatic tumors. These achievements mark a significant advancement in photoacoustic imaging, paving the way for expanded applications in both research and clinical settings. Future developments based on this platform promise to revolutionize minimally invasive diagnostics and monitoring across various medical disciplines.

## Methods

### TUT design

The transducer was designed using the Krimholtz, Leedom, and Matthaei (KLM) model. Simulations were performed using a custom MATLAB-based simulator from a previous paper[66]. Lithium niobite (LNO) was chosen as the piezoelectric material, with a 0–3 ceramic epoxy composite as the first matching layer, and epoxy as the second matching and backing layers. The piezoelectric material size was set to 6 mm × 6 mm to achieve an electrical impedance close to 50 ohms at the TUT's center frequency. The thickness of the LNO was set to half the wavelength of the TUT's center frequency, while the first and second matching layers were set to a quarter of the wavelength at the center frequency. In initial simulations, the thicknesses of each layer were adjusted to maximize bandwidth, resulting in final thicknesses of 19, 30, 94, and 1000 μm for the second matching, first matching, LNO, and backing layers, respectively.

### TUT fabrication

The fabrication of the TUT is schematically illustrated in Supplementary Fig. S15. A 6 mm × 6 mm LNO crystal (Lithium Niobate, Boston Piezo-Optics, USA) was used. Both the front and back sides of the LNO were lapped and polished to a thickness of 94 μm, then 250 nm ITO electrodes were sputtered on both surfaces, and chrome-gold was sputtered on the edges of the crystal. A composite of epoxy (EPOTEK 301, Epoxy Technology, Inc. USA) and ceramic (silicon dioxide, Sigma-Aldrich, USA) was poured and cured for use as the first matching layer. This layer was lapped and polished to a thickness of 30 μm. A second matching layer made of epoxy was poured onto the stack and cured, then processed similarly to achieve a layer thickness of 19 μm. Conductive epoxy was built up on the back corners of the resulting stack for easier electrode connection, and an epoxy backing was poured, cured, and lapped and polished to a thickness of 1 mm. The completed stack was then attached to a brass housing with an inner diameter of 9 mm and an outer diameter of 11 mm, and the front ground electrode was connected to the housing. Finally, a 38 AWG micro-coaxial cable (9438 WH033, Alpha Wire, USA) was connected to the signal and ground electrodes to complete the TUT fabrication. During transducer fabrication, additional processing was repeated at each step until the target specifications were achieved, and as a result, the variations in center frequency and bandwidth across all fabricated devices were maintained within ±10% of the target values.

### Measuring the TUT's acoustic properties

The TUT's two-way acoustic properties were measured by using a pulser/receiver (5073PR, Olympus, Japan) to acquire the pulse-echo signal from a highly reflective crystal and analyzing it. The TUT's characteristics in the frequency domain were analyzed using fast Fourier transform in MATLAB to derive the center frequency and bandwidth.

### Optical property measurement

The optical transmittance of the TUT was measured using a UV–VIS spectrophotometer (S-3100, SCINCO CO., LTD, Repulic of Korea) over a wavelength range from 300 nm to 1100 nm.

### hPAM-TUT PAM probe fabrication

The fiber scanner was fabricated by connecting 28 AWG wires to the +/−XY electrodes of a piezo tube (PT230.94, PI Ceramic GmbH, Germany) and inserting 3D-printed fiber holders into both ends of the tube. A single-mode fiber (P1-460B-FC-5, Thorlabs, USA) was prepared by stripping the cladding at one end, cleaving it to the appropriate length, and then carefully inserting it into the fiber holder. The fiber was aligned centrally and secured using instant adhesive, completing the scanner assembly. The optical system comprised three achromatic lenses (47-652, Edmund Optics Inc., USA), each with a 12 mm focal length and 9 mm diameter. To ensure proper light collimation and maintain coaxial alignment, two lenses were fitted into a 3D-printed lens holder and inserted into a 3D-printed housing. The third lens was positioned at the housing's end to focus the light. To complete the probe assembly, the fiber scanner was inserted into the housing and aligned coaxially with the optical system. The TUT was attached to the distal end of the housing. A water tank was then affixed in a position that aligned the imaging plane with the focal point of the light. The tank was filled with water, sealed with a Teflon film, and capped, finalizing the hPAM-TUT probe fabrication. The acoustic loss of the Teflon film was measured to be 5% as shown in Supplementary Fig. S16.

### hPAM-TUT PAM system

Figure 3 showed a schematic of the hPAM-TUT system. A 532 nm pulsed laser (VPFL-G-10, Spectra-Physics, USA) with a 2 ns pulse duration generates PA signals. The laser beam is directed into a collimator (F230FC-A, Thorlabs, USA) using a mirror with 2-axis angle adjustment. From there, the collimated beam is coupled into a single-mode fiber (P1-460B-FC-5, Thorlabs, USA) at one end of the probe. Light emitted from the fiber cantilever at the distal end of the probe traverses the optical system, TUT, water tank, and film before reaching the target. The overall system operation and data collection was conducted using LabVIEW software (National Instruments, USA). To achieve scanning, the piezo tube actuates the fiber cantilever in a spiral pattern at its resonant frequency. This actuation is driven by an analog sinusoidal voltage generated by a data acquisition board (PCIe6321, National Instruments, USA) and amplified by a piezo amplifier (E-413.20, PI ceramic GmbH, Germany). The resulting PA signals undergo two-stage amplification. First, they are amplified by 50 dB using a custom amplifier (Optc-LNA-X2, Opticho Co., Ltd, Republic of Korea). Subsequently, they pass through a pulser/receiver (5073PR, Olympus, Japan) for secondary amplification of 25 dB and 5 MHz high-pass filtering. The processed signals are then digitized at a 250 MHz sampling rate using a high-speed digitizer (ATS9350, Alazar Technologies Inc, Canada). Synchronization between the laser pulses and data acquisition is managed through the data acquisition board.

### Position sensor-based image reconstruction

Supplementary Fig. S8a presents a schematic of the position data acquisition system. A 532 nm continuous-wave laser was employed, and the DAQ generated a scanning voltage synchronized with the oscilloscope's data acquisition. A custom-built control box supplied power to the position sensor and transmitted the corresponding voltage signals to the oscilloscope. The obtained position data were converted into pixel values using the manufacturer's formula and the maximum pixel size, then rounded to the nearest pixel. Supplementary Fig. S8b illustrates the computation method, which falls into three cases. In the first case, where a pixel corresponds to a single data point, the position data are directly assigned to that pixel. In the second case, when multiple data points correspond to one pixel, the pixel's value is calculated as the average of those data points. In the third case, if no data points correspond to a pixel, an interpolated value from the nearest data points is assigned using a distance-weighted interpolation method. We further validated that the proposed reconstruction approach introduces minimal artifacts by imaging the grid resolution target, as shown in Supplementary Fig. S17. The trajectory needed to be measured only once immediately after probe fabrication, and stable scanning was maintained in subsequent experiments without repeated acquisition of position data.

### Customized triggering setting

The trigger settings were initially determined through a density calculation (laser shots per pixel). Based on a final image size of 512 × 512 pixels, we calculated the time-dependent density and adjusted the PRF to ensure that each pixel received between 1 and 5 laser shots (Fig. 2C).

Subsequently, we conducted a comprehensive simulation in MATLAB, integrating the probe's scanning frequency of 333 Hz with the PRF settings derived from our density calculations. This simulation was designed to validate our approach in a two-dimensional context. The results confirmed that each pixel in the 2D simulation received between 1 and 5 laser shots, in accord with our initial calculations.

### Resolution measurement

The lateral and axial resolutions were measured using a film target. The film target was fixed in water, and images were acquired by moving and positioning the probe using a 5-axis stage. To determine the axial resolution, a single A-line signal was extracted from the acquired data, processed with a Hilbert transform, and then fitted to a Gaussian function. The full width at half maximum (FWHM) of this fitted function was calculated to determine the axial resolution. For lateral resolution measurement, an edge region was selected from this MAP image, and its line profile was extracted. This profile was then fitted to an edge spread function (ESF). The fitted ESF was subsequently used to derive a line spread function (LSF) in the form of a Gaussian function. The FWHM of this LSF was calculated to determine the lateral resolution.

### Imaging a rat's surgically exposed abdominal organs in vivo

All animal experiments were conducted in compliance with the laboratory animal protocol approved by the Institutional Animal Care and Use Committee of Pohang University of Science and Technology (POSTECH-2024-0025). All animals were housed in a specific-pathogen-free (SPF) facility under standard laboratory conditions. They were maintained on a 12-h light/dark cycle at a constant ambient temperature of $22 \pm 2\,°C$ and relative humidity of $50 \pm 10\%$. Food and water were provided ad libitum. The surgically exposed abdominal organs of 8-week-old Sprague Dawley rats were imaged in vivo. Prior to imaging, the rats were anesthetized with 4% isoflurane vaporized by inhalation gas (1.0 L/min flow rate), and anesthesia was maintained with 1.5% isoflurane throughout the procedure. During the experiment, the rat's body temperature was maintained using an infrared lamp. The abdominal cavity of the anesthetized rat was opened minimally using surgical scissors. Ultrasound gel was applied to the tip of the probe to ensure acoustic coupling. The probe was then inserted into the opened abdominal cavity, and images of the rat's stomach, intestine, cecum, and bladder were captured while manually manipulating the probe. Depth encoded and 3D rendered images were processed using 3D PHOVIS[84], a MATLAB-based 3D visualization application.

### Mouse ear monitoring in vivo

In vivo monitoring of ear vasculature changes was conducted on 6-week-old BALB/c nude mice. Prior to imaging, the mice were anesthetized with 2% isoflurane vaporized by inhalation gas (1.0 L/min flow rate), and anesthesia was maintained with 1% isoflurane throughout the procedure. During the experiment, the mouse's body temperature was maintained by an infrared lamp. To ensure acoustic coupling, ultrasound gel was applied to the tip of the probe. For quantitative analysis and accurate monitoring, the probe was fixed to a stage during imaging. The imaging protocol consisted of a 30-s baseline observation period, followed by the IV injection of 5 µg of epinephrine, and then a subsequent 210-s imaging period. Images were acquired at 1.5-s intervals throughout the experiment.

### Mouse ear lymphatic vessel imaging in vivo

In vivo monitoring of the action of Evans blue dye on the mouse's ear lymphatic vessel was conducted on 6-week-old BALB/c nude mice. Prior to imaging, the mice were anesthetized with 2% isoflurane vaporized by inhalation gas (1.0 L/min flow rate), and anesthesia was maintained with 1% isoflurane throughout the procedure. During the experiment, the mouse's body temperature was maintained using an infrared lamp. To ensure acoustic coupling, ultrasound gel was applied to the tip of the probe. For quantitative analysis and accurate monitoring, the probe was fixed to a stage during imaging. Evans blue dye solution (3%, 5 µL) was injected intradermally into the mouse's ear. Images were acquired at 1.5-s intervals throughout the experiment.

### Metastatic tumor model

A metastatic tumor model was established by intrasplenic injection of MC38 mouse colon cancer cells into a 8-week-old female BALB/c nude mouse[85]. After complete anesthesia, a lateral incision was made to expose the spleen. MC38 cells ($1 \times 10^6$) were diluted in Dulbecco's phosphate-buffered saline and slowly injected into the spleen. To prevent cell leakage, the injection point was softly pressed with a cotton swab after injection. Following ligation of the splenic vessels, the spleen was removed and the incision was sutured. All surgeries were performed with sterile instruments and conditions.

### Mouse metastatic tumor imaging in vivo

In vivo imaging of the established abdominal metastatic tumor was conducted 14 days after tumor cell inoculation. Prior to imaging, the mice were anesthetized with 2% isoflurane vaporized by inhalation gas (1.0 L/min flow rate), and anesthesia was maintained with 1% isoflurane throughout the procedure. The mouse's abdominal cavity was opened using scissors, ultrasound gel was applied to the probe, and the imaging was performed in a handheld manner.

### Quantitative analysis of tumor vessel metrics

Supplementary Fig. S14a shows a PA MAP image of the tumor region. A threshold was applied to this image to generate a vascular mask (Supplementary Fig. S14b). Supplementary Fig. S14c displays the skeleton extracted from the vascular mask of tumor region. Supplementary Fig. S14d–f shows PA MAP image, vascular mask, and vascular skeleton of normal region respectively. Several vessel metrics were derived from the vascular mask and skeleton. The Vascular Area Density (VAD)—defined as the ratio of the vascular area to the imaging area—was calculated by dividing the masked vascular area by the total imaging area. The Vessel Complexity Index (VCI) was computed as the square of the vascular perimeter divided by the product of the vascular area and $4\pi$, quantifying vessel irregularity. The Vessel Skeleton Density (VSD) was defined as the total skeleton length normalized by the imaging area, while the Vessel Diameter Index (VDI) was defined as the vascular area divided by the skeleton length. All measurements were converted from pixel units to physical dimensions using the calibrated pixel size of the imaging system.

### Reporting summary

Further information on research design is available in the Nature Portfolio Reporting Summary linked to this article.

## Data availability

All data supporting the findings of this study are available within the paper and its Supplementary Information. Additional data are available from the corresponding author upon request and responses to data requests will be provided within one month.

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

## Acknowledgements

This research was supported by following sources: National Research Foundation of Korea (NRF) funded by the Ministry of Science and ICT (MSIT) (RS-2023-NR077260, C.K.; no. RS-2025-00558651, 2710079568, J.K.; no. RS-2023-00212322, 2710076852, S.C.), the Basic Science Research Program through the Ministry of Education (RS-2020-NR049599, C.K.), the Commercialization Promotion Agency for R&D Outcomes (COMPA) funded by the Ministry of Science and ICT (MSIT) (RS-2025-02304660, C.K.), BK21 FOUR program (C.K.), Glocal 30 University Project (C.K.), the Hyundai Motor Chung Mong-Koo Foundation (M.H.)

## Author contributions

C.K., M.H., and J.K. conceived and designed the study. M.H. and J.K. constructed the overall system. M.H. fabricated the TUT, with assistance from S.C., D.H., and M.K. M.H. and J.A. developed the program used for the experiments. M.H., J.K., J.L., E.P., J.Y.K., Y.K., Y.J.A., H.H.K., and W.J.K. performed experiments and acquired data. M.H., J.K. and S.C visualized the data. C.K. supervised the project. M.H., J.K., and C.K. wrote the manuscript.

## Competing interests

C.K. and J.A. have financial interests in Opticho Co., Ltd., which did not support this work. J.K. and SC were previously affiliated with Opticho Co., Ltd., but are no longer associated with the company and have no competing interests related to this work. All other authors declare no competing interests.
