## [Transparent Peer Review file · Nature Communications]

A handheld photoacoustic microscopic probe integrating a transparent ultrasound transducer and a fiber scanner

Corresponding Author: Professor Chulhong Kim

Version 0:

Reviewer comments:

Reviewer #1

(Remarks to the Author)

Overall Summary

This manuscript introduces hPAM-TUT, a hand-held optical-resolution photoacoustic microscope that marries a resonant fiber scanner with a high-frequency transparent ultrasound transducer. The probe is remarkably compact (17 mm \varnothing \times 90 mm length) yet maintains a 7 μ m lateral / 47 μ m axial resolution, a 2.6 mm field of view and a 1.5 s volume-rate. In vivo studies—rat abdominal organs, epinephrine-induced vasoconstriction, Evans-blue lymphangiography and early metastatic lesions—provide convincing proof-of-principle. The ability to coaxially align optics and acoustics without a beam combiner represents a clear technical advance and positions the device well for translational applications in dermatology, oncology and intra-operative guidance. I am enthusiastic about the work and recommend the suggestions below to strengthen the manuscript.

Major Comments

1. All experimental demonstrations are for superficial tissues ($< \approx 400 \mu$ m). Please add quantitative SNR-versus-depth and lateral resolution curves in tissue-mimicking phantoms or in vivo to document practical imaging depth.
2. The customised PRF scheme equalises fluence spatially, but pulse energy, spot size and resulting fluence at the tissue are not reported. Provide these numbers and compare with ANSI MPE limits, commenting on photothermal safety for prospective clinical use.
3. Several biological examples use small cohorts. Adding confidence intervals and statistical tests would bolster the biological claims (e.g., vessel-density drop after epinephrine).
4. While the hardware is hand-held, dynamic studies appear to have been performed with the probe fixed to stages. A brief discussion of motion-correction capability would confirm clinical practicality.
5. Better elaborate the novelty of this handheld work vs. other handheld PA imaging systems, for example, from Lihong Wang and Lidai Wang's labs. A comparison table of the key imaging performance would be helpful.

Minor Comments

1. Please list pulse energy at the fiber tip, spot diameter on tissue and resultant fluence.
2. Quantify the SNR variation across the FOV and consider normalising the MAP images or reporting a per-pixel SNR map.
3. How about the long-term drift of the 333 Hz resonance (e.g., over 1 h of continuous operation) and any temperature dependence, as both impact pixel mapping accuracy.
4. How about the position-sensor's accuracy?
5. The simulated fill factor reaches $>95\%$ above 250 Hz (Supplementary Fig. S3), can this be experimentally confirmed with different scanning configurations?
6. The simulated acoustic intensity (Supplementary Fig. S2) suggests a 4 mm lateral uniform zone at 8 mm depth. An

experimental beam map (needle hydrophone scan or pulse-echo wire test) would confirm coverage and quantify acceptance-angle limits for oblique tissues.

7. The 50 μm Teflon membrane imposes any acoustic loss at ~ 37 MHz?

8. Indicate typical yield (functional devices per wafer) and variability in centre frequency/bandwidth to reassure future adopters of the TUTs.

9. Vessel, lymphatic and tumour masks rely on manual thresholds (Supplementary Texts 3–5). Discuss the impact of mask threshold on the analysis.

10. Provide total probe mass and a photograph of the probe in a gloved hand to convey clinical handling comfort.

Overall, the authors present a technically elegant and potentially impactful advance. Addressing the points above will further highlight the system's translational value.

Reviewer #2

(Remarks to the Author)

This manuscript presents a handheld photoacoustic microscopy probe (hPAM-TUT) integrating a transparent ultrasound transducer (TUT) and a fiber scanner. The system used a variable repetition rate to achieve uniform sampling, which features a high lateral and axial resolution, and demonstrates imaging applications in various animal models. The overall design is clear, and the experimental results are convincing, showcasing good engineering practicality and publication potential, however, the manuscript still lacks details in some experimental methods and should be major revised before it can be accepted.

1. The imaging probe has a diameter of 17 mm. At this relatively large size, fiber resonant scanning may not be the optimal scanning method compared to MEMS mirrors, as it can introduce significant nonlinear effects and instability due to temperature variations and mounting structure issues. Moreover, recent studies have demonstrated that forward-viewing photoacoustic endoscopic probes based on fiber scanning can already achieve diameters smaller than 4 mm (Ke, D. et al. Miniature fiber scanning probe for flexible forward-view photoacoustic endoscopy. Applied Physics Letters 122 (2023)). How did the authors make this design decision, and what was the reason for not using a MEMS mirror?

2. The authors used a time-divided PRF strategy to address the issue of excessive sampling density in the central region. It is a good approach to avoid heat accumulation at the center. However, the manuscript and supplementary materials do not include comparative imaging data between constant and variable repetition rate modes. Please provide the corresponding comparison data to illustrate the effect on image quality.

3. In the section concerning variable repetition rates, the authors should add a quantitative comparative experimental discussion on the relationship between the variable repetition rate and the imaging speed.

4. The image reconstruction using an external PSD trajectory, but the description is too brief. The authors should describe the reconstruction process in more detail in the Methods section.

5. No spiral artifacts caused by the fiber scanning trajectory were observed in the image. What methods did the authors use. The authors may need to add imaging results of structured samples, such as a UASF 1951, to demonstrate that the system does not introduce distortion.

6. The manuscript provides limited description of the vessel extraction and quantification methods (VAD, VDI, VCI, VSD), it is recommended to provide more details in Methods section.

7. What is the temporal stability of the probe? Is trajectory calibration required before each experiment?

8. Can fiber resonant scanning be used in clinical applications in the future? Is this scanning method sufficient and reliable enough? How do the authors view this issue?

Version 1:

Reviewer comments:

Reviewer #1

(Remarks to the Author)

The authors have addressed my comments in full. Thanks.

Reviewer #2

(Remarks to the Author)

I do not fully agree with the authors' response to Comment 1. The system presented in this manuscript employs a non-coaxial configuration of optical and ultrasonic paths, where only the optical beam is scanned. Since a transparent transducer is used and the optical scanning component is positioned behind the transducer, the issue of MEMS immersion in water

does not apply to this design. Furthermore, the stability of MEMS scanners is not necessarily inferior to that of fiber scanners - in fact, MEMS devices may even offer better stability. Therefore, I suggest that the authors revisit and discuss this point more thoroughly.

Reviewer #1:

This manuscript introduces hPAM-TUT, a hand-held optical-resolution photoacoustic microscope that marries a resonant fiber scanner with a high-frequency transparent ultrasound transducer. The probe is remarkably compact (17 mm \varnothing \times 90 mm length) yet maintains a 7 μ m lateral / 47 μ m axial resolution, a 2.6 mm field of view and a 1.5 s volume-rate. In vivo studies—rat abdominal organs, epinephrine-induced vasoconstriction, Evans-blue lymphangiography and early metastatic lesions—provide convincing proof-of-principle. The ability to coaxially align optics and acoustics without a beam combiner represents a clear technical advance and positions the device well for translational applications in dermatology, oncology and intra-operative guidance. I am enthusiastic about the work and recommend the suggestions below to strengthen the manuscript.

Reply: *Thank you for the positive comments.*

Major Comments

Comment 1: All experimental demonstrations are for superficial tissues ($< \approx 400 \mu$ m). Please add quantitative SNR-versus-depth and lateral resolution curves in tissue-mimicking phantoms or in vivo to document practical imaging depth.

Reply: *We appreciate the reviewer's comment. As suggested, we measured the lateral resolution and SNR at different depths in both tissue-mimicking phantoms and water, and added the results to Supplementary Fig. S9. The corresponding description has been included in Supplementary Text 3.*

Results

Page 8 Line 208:

“Depth-dependent resolution and SNR curves measured in water and tissue-mimicking phantoms are presented in Supplementary Text 3 and Fig. S9.”

Supplementary Text 3. Imaging performance test in water and tissue-mimicking phantoms.

Page 22 Line 177:

“To evaluate the imaging performance in tissue, we measured the lateral resolution and signal-to-noise ratio (SNR) at various depths in both water and tissue-mimicking phantoms (Intralipid 0.5% and Intralipid 1%) (Fig. S9). A printed film served as the imaging target. The edge of the printed film was imaged, and the line spread function (LSF) was derived from the edge spread function (ESF). The full width at half maximum (FWHM) of the LSF was calculated to determine the lateral resolution. The highest lateral resolutions were 11.5 μ m, 14.7 μ m, and 23.2 μ m in water, Intralipid 0.5%, and Intralipid 1%, respectively. SNR was measured from the same target, with maximum values of 38.2 dB, 37.6 dB, and 35.0 dB, which gradually decreased with imaging depth to 32.9 dB, 23.9 dB, and 19.2 dB in water, Intralipid 0.5%, and Intralipid 1%, respectively. These results indicate that both resolution and SNR degrade with increasing scattering, as expected in tissue-mimicking media. Beyond $\sim 800 \mu$ m, the resolution degrades and the SNR decreases sharply, limiting the maximum imaging depth to $\sim 800 \mu$ m.”

Supplementary Fig. S9. Lateral resolution and SNR curves measured in water and tissue-mimicking phantoms. (a) Lateral resolution curves, and (b) SNR curves measured in water, intralipid-0.5%, and intralipid-1.0%.

Comment 2. The customised PRF scheme equalises fluence spatially, but pulse energy, spot size and resulting fluence at the tissue are not reported. Provide these numbers and compare with ANSI MPE limits, commenting on photothermal safety for prospective clinical use.

Reply: Following description has been incorporated in Results section and Supplementary Text 2.

Results

Page 6 Line 183:

“Details on laser safety are provided in Supplementary Text 2.”

Supplementary Text 2. Laser safety in in vivo test

Page 22 Line 168:

“In all in vivo experiments of this study, the laser pulse energy was 600 nJ. The focal point of the system was formed about 250 µm beneath the film in the water tank; however, during actual imaging with tissue contact, the film was slightly compressed, and the focus was formed at approximately 300 µm within the tissue. Based on an NA of 0.18, the beam spot diameter at the tissue surface was estimated to be ~82 µm, corresponding to an energy density of ~11.3 mJ/cm². This value is below the ANSI MPE” limit for skin (20 mJ/cm²), indicating a high level of photothermal safety for clinical applications.”

Comment 3. Several biological examples use small cohorts. Adding confidence intervals and statistical tests would bolster the biological claims (e.g., vessel-density drop after epinephrine).

Reply: To increase statistical significance, we conducted additional epinephrine experiments as suggested. The in vivo results were revised accordingly, and the manuscript was updated to reflect this change.

Results

Page 12 Line 250:

“This experiment was repeated in five mice to ensure reproducibility”,

Page 12 Line 256:

“The vessel density decreases by approximately 60% overall, with the microvessels showing up to 68% constriction and the macrovessels exhibiting a maximum reduction of about 26%. These findings demonstrate the strong vasoconstrictive response induced by epinephrine, mediated by alpha-adrenergic receptors in vascular smooth muscle. Arterioles, being thinner and containing more smooth muscle than venules, contract more significantly, as noted in earlier reports 70-72. Additionally, the PA signals decrease by up to 6.5 dB in the microvessels but only 3.3 dB in the macrovessels”.

Fig. 5. Epinephrine-induced vasoconstriction in the vasculature of a mouse's ear. (a) PA MAP images of the vasculature at pre-injection and 30-, 60-, and 90-seconds post-epinephrine injection (**Supplementary Movie S3**). (b) Quantitative analysis of changes in vessel density and PA amplitude of total, macro (>100 μm), and micro (<100 μm) vessels across the entire region. **Data are presented as mean \pm 95% CI (n = 5).** (c) Magnified images of the white box in (a). (d) Quantitative analysis of change in the vascular area in the white dashed box in (c). **CI, confidence interval**

Comment 4. While the hardware is hand-held, dynamic studies appear to have been performed with the probe fixed to stages. A brief discussion of motion-correction capability would confirm clinical practicality.

Reply: *Following information has been added to the Discussion section.*

Discussion

Page 19 Line 421:

"Fourth, current dynamic experiments are performed by mounting the probe on a fixed stage to precisely quantify microvascular contraction. However, motion artifacts caused by hand tremor or patient movement are unavoidable in future clinical applications. To minimize these issues, from a hardware perspective, increasing the image frame rate is crucial for acquiring less distorted images. This can be achieved by increasing the resonant frequency through increasing the mechanical stiffness of the optical fiber, for example, by enlarging the cladding diameter or applying a metal coating, thereby enabling faster scanning. Such hardware-based approaches are expected to effectively suppress motion artifacts occurring in clinical settings and provide stable imaging performance."

Comment 5. Better elaborate the novelty of this handheld work vs. other handheld PA imaging systems, for example, from Lihong Wang and Lidai Wang's labs. A comparison table of the key imaging performance would be helpful.

Reply: Thanks for the suggestion. Following table has been added in the Supplementary Table S1.

Introduction

Page 2 Line 77:

“A comprehensive comparison with other handheld photoacoustic probe studies is provided in Supplementary Table S1.”

	Scanner	Optical scanning	Size	FOV	Imaging speed	Resolution [μm]		Application
K. Park et al. (2017)⁴⁶	MEMS	X	\varnothing 17 mm x 120 mm	$2 \times 2 \text{ mm}^2$	20s / volume	L : 12	A : 30	Human mole
L. Lin et al. (2017)⁴⁷	MEMS	O	80 x 115 x 150 mm ³	$2.5 \times 2.0 \text{ mm}^2$	0.5s / volume	L : 5	A : 26	Mouse ear, Human mole
Q. Chen et al. (2018)⁴⁵	MEMS	O	22 x 30 x 13 mm ³	$2 \times 2 \text{ mm}^2$	5s / volume	L : 3.8	A : 104	Rat internal organs, Human oral cavity
W. Zhang et al. (2020)³⁹	MEMS	O	\varnothing 12 mm	\varnothing 2.4 mm	4s / volume	L : 18.2 137.4	A :	Human oral cavity
W. Qin et al. (2021)³⁷	Galvanometer + rotator	O	NM	\varnothing 10 mm	5s / volume	L : 15	A : 120	Rhesus cerebral cortex
J. Chen et al. (2022)³⁶	Galvanometer + MEMS	X	59 x 30 x 44 mm ³	$1.7 \times 5 \text{ mm}^2$ $1.7 \times 2 \text{ mm}^2$	0.5s / volume 0.22s / volume	L : 6.2	A : 39	Rat internal organs, mouse brain stroke, human lip
D. Ke et al. (2023)³⁵	Fiber scanner	O	\varnothing 5 mm	\varnothing 3 mm	2s / volume	L : 15.6	A : 168	Rat stomach, intestine
This work	Fiber scanner	O	\varnothing 17 mm x 90 mm	\varnothing 2.6 mm	1.5s / volume	L : 7	A : 47	Rat internal organs, mouse lymphatic vessel, mouse metastatic tumor

Supplementary Table. S1. Comprehensive comparison with other handheld photoacoustic probes. FOV, field of view; L, lateral resolution; and A, axial resolution.

Minor Comments

Comment 1. Please list pulse energy at the fiber tip, spot diameter on tissue and resultant fluence.

Reply: Done (Major comment #2)

Comment 2. Quantify the SNR variation across the FOV and consider normalising the MAP images or reporting a per-pixel SNR map.

Reply: We additionally imaged black tape to provide a pixel-wise signal intensity map in the MAP images, which has been included in Supplementary Fig. S10.

Results

Page 8 Line 210:

"In addition, we evaluated the uniformity of the optical excitation and ultrasonic receiving regions within the FOV by imaging black tape to generate a SNR map. The difference between the strongest and weakest signals in the image was within 1.85 dB (Supplementary Fig.S10)."

Supplementary Fig. S10. Signal intensity map across the FOV.

Comment 3. How about the long-term drift of the 333 Hz resonance (e.g., over 1 h of continuous operation) and any temperature dependence, as both impact pixel mapping accuracy.

Reply: *We evaluated the long-term pixel mapping accuracy. The initial trajectory was recorded using a position sensor, followed by continuous operation for one hour. Subsequent measurements showed negligible deviation in the trajectory. Although precise control of temperature variations was limited in our current experimental setup, we confirmed that long-term thermal stability was maintained under typical laboratory conditions. This result has been supplemented in the Results section.*

Results

Page 8 Line 213:

"The scanning accuracy of the piezoelectric tube was maintained even during long-term operation exceeding 1 hour (Supplementary Fig. S11)."

Supplementary Fig. S11. Scanning trajectory measured in position sensor using 333 Hz. Scanning trajectory (a) before and (b) after continuous 1 hour operation.

Comment 4. How about the position-sensor's accuracy?

Reply: *The position sensor used in this study was the PDP90A model from Thorlabs, which has a positional resolution of 0.75 μm according to the manufacturer's specifications. Therefore, it is considered sufficiently suitable for evaluating pixel mapping accuracy.*

Comment 5. The simulated fill factor reaches >95 % above 250 Hz (Supplementary Fig. S3), can this be experimentally confirmed with different scanning configurations?

Reply: *As the scanning range of the fiber scanner rapidly decreases when deviating from the resonance frequency, a fill factor has limited practical significance. However, measurements at several frequencies near the resonance frequency of 333 Hz showed that the fill factor remained above 95% in all cases. This result has been added in Supplementary Fig. S4 and Results section*

Results

Page 6 Line 157:

"This simulation has been experimentally shown in Supplementary Fig. S4."

Supplementary Fig. S4. Scanning trajectory measured using various scanning frequencies. Scanning trajectory measured using (a) 303 Hz, (b) 313 Hz, (c) 323 Hz, (d) 343 Hz, (e) 353 Hz, (f) 363 Hz.

Comment 6. The simulated acoustic intensity (Supplementary Fig. S2) suggests a 4 mm lateral uniform zone at 8 mm depth. An experimental beam map (needle hydrophone scan or pulse-echo wire test) would confirm coverage and quantify acceptance-angle limits for oblique tissues.

Reply: *To experimentally acquire acoustic pressure maps and verify coverage, hydrophone measurements were conducted. Although the measured results differed slightly from the simulation, it was confirmed that signals could be reliably detected within the system’s field of view. The corresponding data have been added to Supplementary Material S2, and the Results section has been updated accordingly.*

Results

Page 4 Line 138:

“Yet, measured acoustic field showed that TUT could receive signal evenly in about 2.6 mm range (Supplementary Fig. S2)”

Supplementary Fig. S2. Simulated and measured acoustic intensity field. (a) Simulated acoustic field. (b) Measured acoustic field.

Comment 7. The 50 μm Teflon membrane imposes any acoustic loss at ~ 37 MHz?

Reply: To quantify the acoustic loss of the Teflon membrane, we compared PA signals with and without the membrane, and observed a signal reduction of approximately 5%. This has been added to the Materials and Methods section.

Materials and Methods

Page 21 Line 492:

“The acoustic loss of the Teflon film was measured to be 5% as shown in Supplementary Fig. S16.”

Supplementary Fig. S16. Photoacoustic signal comparison with and without Teflon film.

Comment 8. Indicate typical yield (functional devices per wafer) and variability in centre frequency/bandwidth to reassure future adopters of the TUTs.

Reply: *The TUTs used in this study were fabricated through individually optimized processes rather than wafer-scale mass production; therefore, the conventional concept of yield is not applicable. A clarification has been added to the Materials & Methods section.*

Materials and Methods

Page 20 Line 466:

"During transducer fabrication, additional processing was repeated at each step until the target specifications were achieved, and as a result, the variations in center frequency and bandwidth across all fabricated devices were maintained within $\pm 10\%$ of the target values."

Comment 9. Vessel, lymphatic and tumour masks rely on manual thresholds (Supplementary Texts 3–5). Discuss the impact of mask threshold on the analysis.

Reply: We analyzed the histogram of the image and manually set the threshold near a peak distinguishable from the noise peak. It was confirmed that the overall trend of the results does not change significantly within $\pm 5\%$ of the chosen threshold. The graphs using the selected threshold and thresholds within the $\pm 5\%$ range are shown below.

Fig. R1. Impact of threshold on vessel density analysis. CI, confidence interval

Comment 10. Provide total probe mass and a photograph of the probe in a gloved hand to convey clinical handling comfort.

Reply: Following the suggestion, a photograph of the probe in operation while wearing gloves is attached in Fig. 3(a). The probe's mass was measured as 11 g, which is also reflected in the Results section.

Results

Page 8 Line 193:

"The entire probe weighed 11 g, which was sufficiently small and light to allow handheld imaging."

Fig. 3. Schematic and performance of a handheld PAM probe with the TUT and fiber scanner (hPAM-TUT). (a) Schematic and photograph of the hPAM-TUT. (b) Phantom and associated PAM image. (c) Lateral and axial resolutions. DAQ, data acquisition device; DGT, digitizer; PR, pulsar/receiver; P AMP, piezo amplifier; AMP, amplifier; M, mirror; Col, collimator; PT, piezo tube; FC, fiber cantilever; WT, water tank; ESF, edge spread function; and LSF, line spread function.

Overall, the authors present a technically elegant and potentially impactful advance. Addressing the points above will further highlight the system's translational value.

Reply: Thank you again.

Reviewer #2

This manuscript presents a handheld photoacoustic microscopy probe (hPAM-TUT) integrating a transparent ultrasound transducer (TUT) and a fiber scanner. The system used a variable repetition rate to achieve uniform sampling, which features a high lateral and axial resolution, and demonstrates imaging applications in various animal models. The overall design is clear, and the experimental results are convincing, showcasing good engineering practicality and publication potential, however, the manuscript still lacks details in some experimental methods and should be major revised before it can be accepted.

Reply: Thank you for the positive comments.

Comment 1. The imaging probe has a diameter of 17 mm. At this relatively large size, fiber resonant scanning may not be the optimal scanning method compared to MEMS mirrors, as it can introduce significant nonlinear effects and instability due to temperature variations and mounting structure issues. Moreover, recent studies have demonstrated that forward-viewing photoacoustic endoscopic probes based on fiber scanning can already achieve diameters smaller than 4 mm (Ke, D. et al. Miniature fiber scanning probe for flexible forward-view photoacoustic endoscopy. Applied Physics Letters 122 (2023)). How did the authors make this design decision, and what was the reason for not using a MEMS mirror?

Reply: We appreciate the reviewer's insightful comment. The reasons for choosing a fiber scanner over MEMS in the handheld PAM probe are as follows. First, MEMS-based handheld systems typically combine optics and ultrasound coaxially and scan both beams simultaneously using MEMS. In such cases, the entire ultrasound path before and after the MEMS must be immersed in water, which significantly reduces the scanning speed due to water resistance or requires a larger MEMS to compensate. Moreover, the system design necessitating full immersion inevitably increases the overall size of the device. Second, nonlinear scanning effects mainly occur in MEMS, whereas fiber scanners exhibit linear scanning characteristics with high reproducibility, and they are also considerably less sensitive to temperature variations compared to MEMS. Third, in the case of the 4 mm endoscope reported by Ke. D et al., a forward-viewing structure with an opaque transducer requires a beam combiner and extends the ultrasound propagation path. Consequently, high-frequency ultrasound cannot be effectively used due to rapid attenuation, restricting operation to lower frequencies and resulting in an axial resolution of 168 μm (compared to 47 μm in our system). This directly impacts the ability to distinguish vascular structures in the axial direction, as shown in Fig. 6. In contrast, our system resolves the trade-offs among high speed, miniaturization, and high spatial resolution through the integration of a transparent transducer and fiber scanner. The current 17 mm diameter reflects an optimal arrangement of the main components, with further miniaturization possible in the future. This information has been added to the introduction.

Materials and Methods

Page 2 Line 81:

“The reasons for choosing a fiber scanner over MEMS in our handheld PAM probe are as follows. First, MEMS scanners require the entire ultrasound propagation path to be immersed in water, which significantly reduces scanning speed; compensating for this necessitates larger MEMS devices, thereby increasing the overall system size. Second, MEMS scanners often suffer from nonlinear scanning effects, whereas fiber scanners provide highly reproducible linear scanning characteristics and are far less sensitive to temperature variations. Third, in another study³⁵, a forward-viewing structure with an opaque transducer required a beam combiner and extended the acoustic propagation path. As a result, high-frequency ultrasound could not be effectively used due to rapid attenuation, and the operating frequency was limited to the low-frequency range, yielding an axial resolution of only 168 μm (compared with 47 μm in our system). This limitation reduces the ability to resolve vascular structures in the axial direction. By contrast, our system integrates a transparent transducer with a fiber scanner to resolve the trade-offs among high speed, miniaturization, and high spatial resolution.”

Comment 2. The authors used a time-divided PRF strategy to address the issue of excessive sampling density in the central region. It is a good approach to avoid heat accumulation at the center. However, the manuscript and supplementary materials do not include comparative imaging data between constant and variable repetition rate modes. Please provide the corresponding comparison data to illustrate the effect on image quality.

Reply: Following the suggestion, we conducted a comparative experiment between fixed PRF mode and variable PRF mode. Using printed targets, we confirmed that both modes produced similar image quality. However, in fixed PRF mode, heat accumulation occurred in the central region, damaging the sample. A related explanation has been added to the Results Section, and comparative data is included in Supplementary Fig. S6.

Results

Page 6 Line 182:

“Comparative imaging results between the fixed PRF mode and the variable PRF mode are presented in Supplementary Fig. S6”

Supplementary Fig. S6. Comparative imaging results between fixed PRF mode and variable PRF mode. Image of a film target using (a) fixed PRF mode and (b) variable PRF mode.

Comment 3. In the section concerning variable repetition rates, the authors should add a quantitative comparative experimental discussion on the relationship between the variable repetition rate and the imaging speed.

Reply: We have added Supplementary Fig. S5, which shows the relationship between time and fill factor when using a fixed PRF. As mentioned in the main text, operating at a high fixed PRF leads to heat accumulation and energy concentration issues, whereas reducing the PRF to mitigate these issues results in a longer imaging time. This quantitative comparison highlights the trade-off between PRF, energy deposition, and imaging speed. Related explanation is added on Results section.

Supplementary Fig. S5. Imaging time versus fill factor graph using fixed PRF.

Results

Page 6 Line 176:

“A lower fixed PRF results in a slower imaging speed (Supplementary Fig. S5).”

Comment 4. The image reconstruction using an external PSD trajectory, but the description is too brief. The authors should describe the reconstruction process in more detail in the Methods section.

Reply: *It was briefly included in Supplementary Text 4. We have added further details in connection with other comments and moved it to the main text under the Materials and Methods section.*

Results

Page 8 Line 203:

“(details in Materials & Methods section and Supplementary Fig. S8)”

Materials and Methods

Page 21 Line 511:

“Position sensor-based image reconstruction

Supplementary Fig. S8(a) presents a schematic of the position data acquisition system. A 532 nm continuous-wave laser was employed, and the DAQ generated a scanning voltage synchronized with the oscilloscope’s data acquisition. A custom-built control box supplied power to the position sensor and transmitted the corresponding voltage signals to the oscilloscope. The obtained position data were converted into pixel values using the manufacturer’s formula and the maximum pixel size, then rounded to the nearest pixel. Supplementary Fig. S8(b) illustrates the computation method, which falls into three cases. In the first case, where a pixel corresponds to a single data point, the position data are directly assigned to that pixel. In the second case, when multiple data points correspond to one pixel, the pixel’s value is calculated as the average of those data points. In the third case, if no data points correspond to a pixel, an interpolated value from the nearest data points is assigned using a distance-weighted interpolation method.”

Comment 5. No spiral artifacts caused by the fiber scanning trajectory were observed in the image. What methods did the authors use. The authors may need to add imaging results of structured samples, such as a UASF 1951, to demonstrate that the system does not introduce distortion.

Reply: Our system leveraged the high reproducibility of the fiber scanner and reconstructed images using a position sensor, during which no spiral artifacts were observed. To validate this, we imaged a grid resolution target, and the results are included in Materials and Methods section and Fig. S17.

Materials and Methods

Page 22 Line 522:

“We further validated that the proposed reconstruction approach introduces minimal artifacts by imaging the grid resolution target, as shown in Supplementary Fig. S17.”

Supplementary Fig. S17. Grid target image

Comment 6. The manuscript provides limited description of the vessel extraction and quantification methods (VAD, VDI, VCI, VSD), it is recommended to provide more details in Methods section.

Reply: *This content was originally presented in Supplementary Text 3, but based on your suggestion, we have provided additional details and incorporated it into the Methods section of the main text.*

Materials and Methods

Page 23 Line 591:

“Quantitative analysis of tumor vessel metrics

Supplementary Fig. S14(a) shows a PA MAP image of the tumor region. A threshold was applied to this image to generate a vascular mask (Supplementary Fig. S14(b)). Supplementary Fig. S14(c) displays the skeleton extracted from the vascular mask of tumor region. Supplementary Fig. S14(d), S14(e) and S14(f) shows PA MAP image, vascular mask, and vascular skeleton of normal region respectively. Several vessel metrics were derived from the vascular mask and skeleton. The Vascular Area Density (VAD)—defined as the ratio of the vascular area to the imaging area—was calculated by dividing the masked vascular area by the total imaging area. The Vessel Complexity Index (VCI) was computed as the square of the vascular perimeter divided by the product of the vascular area and 4π , quantifying vessel irregularity. The Vessel Skeleton Density (VSD) was defined as the total skeleton length normalized by the imaging area, while the Vessel Diameter Index (VDI) was defined as the vascular area divided by the skeleton length. All measurements were converted from pixel units to physical dimensions using the calibrated pixel size of the imaging system.”

Comment 7. What is the temporal stability of the probe? Is trajectory calibration required before each experiment?

Reply: *The long-term stability of the fiber scanner is addressed in our response to Reviewer #1, Minor Comment 3. No additional trajectory correction was required for each experiment; once the resonant frequency was set and the trajectory measured immediately after probe fabrication, stable scanning was consistently maintained. This clarification has been incorporated into the Materials and Methods section.*

Materials and Methods

Page 22 Line 524:

“The trajectory needed to be measured only once immediately after probe fabrication, and stable scanning was maintained in subsequent experiments without repeated acquisition of position data.”

Comment 8. Can fiber resonant scanning be used in clinical applications in the future? Is this scanning method sufficient and reliable enough? How do the authors view this issue?

Reply: *Fiber resonant scanning enables high-speed imaging, does not emit electromagnetic radiation, and is inherently resistant to external electromagnetic interference. In addition, as long as the fiber remains mechanically intact, its resonant trajectory remains highly stable over time, (as shown in long term operation experiment) providing clinical advantages over MEMS-based scanners. Its simple actuation mechanism also makes it well-suited for probe miniaturization and lightweight design. However, unlike MEMS, fiber scanning has so far been limited to optical-only imaging, such as OCT, because it can scan only light. For photoacoustic imaging, ultrasound detection is required, but conventional opaque transducers necessitate the use of beam combiners, which extend the acoustic path and restrict the system to low-frequency transducers. In this study, we overcame these limitations through integration with a TUT. The TUT enables coaxial collection of PA signals with high axial resolution, allowing simultaneous achievement of miniaturization, high-speed imaging, and superior axial resolution. Thus, the proposed TUT–fiber scanner hybrid provides a breakthrough solution with strong potential for clinical translation. Insights on this have been added at the end of the Discussion section.*

Materials and Methods

Page 19 Line 432:

“In particular, fiber resonant scanning offers high-speed, electromagnetically safe, and long-term stable performance, making it more clinically viable than MEMS-based scanners.”

Reviewer #1:

The authors have addressed my comments in full. Thanks.

Reply: *We sincerely thank the reviewer for their thoughtful comments and constructive feedback. Your suggestions have greatly contributed to improving the clarity and quality of our study, and we truly appreciate the time and effort you invested in reviewing our work.*

Reviewer #2

I do not fully agree with the authors' response to Comment 1. The system presented in this manuscript employs a non-coaxial configuration of optical and ultrasonic paths, where only the optical beam is scanned. Since a transparent transducer is used and the optical scanning component is positioned behind the transducer, the issue of MEMS immersion in water does not apply to this design. Furthermore, the stability of MEMS scanners is not necessarily inferior to that of fiber scanners - in fact, MEMS devices may even offer better stability. Therefore, I suggest that the authors revisit and discuss this point more thoroughly.

***Reply:** Thank you for your insightful comment. We sincerely apologize for the misunderstanding of your previous remark, which resulted in an inaccurate response in our earlier revision. We fully acknowledge that MEMS mirror scanning systems have demonstrated excellent performance and stability in many handheld and compact photoacoustic systems, and we recognize that MEMS remains a highly competitive option for handheld imaging probes. Indeed, we are aware that recent studies have demonstrated excellent forward-view probes utilizing MEMS^{1,2}. While MEMS mirrors are a strong choice, we selected a piezo tube based fiber scanner for our forward-viewing handheld probe for practical and technical reasons. In a forward-view configuration, using a MEMS mirror typically requires at least one reflection to fold the optical path. Such folding increases the optical path length and limits the effective optical numerical aperture, which can constrain the achievable lateral resolution. In addition, including reflective optics and other components within the limited probe space increases mechanical and optical complexity, making alignment and fabrication more challenging.*

In contrast, a fiber resonant scanner allows direct forward illumination without optical folding, resulting in a simpler structure, a shorter optical path, and sufficient scanning stability. Additionally, the fiber scanner can operate at relatively low driving voltages compared to electrostatic MEMS, enabling stable and fast scanning. This is a practical advantage well suited for handheld biomedical research applications. Furthermore, we are also planning future research on the miniaturization of this probe. As the fiber scanner has already been successfully applied in 1-mm-diameter endoscopic systems, this proven potential for miniaturization, suggesting our current design can be directly scaled down, was also one of our key considerations in choosing the fiber scanner. For these reasons, we chose the fiber scanner, and the manuscript has been revised accordingly and references are added. We sincerely appreciate your valuable feedback.

Introduction

Page 2 Line 84:

“Fiber scanners are inherently well-suited for forward-view probe configurations. They achieve wide-range scanning using minimal optical components, which significantly reduces system complexity and simplifies fabrication within the confined probe space. Additionally, their low driving voltage provides enhanced safety for hand-held biomedical applications.”

- 1 Liang, X. *et al.* Multi-scenario photoacoustic endoscopy for in vivo functional imaging. *Photoacoustics* **45**, 100750 (2025).
<https://doi.org/https://doi.org/10.1016/j.pacs.2025.100750>

- 2 Li, L. *et al.* Double spiral resonant MEMS scanning for ultra-high-speed miniaturized optical microscopy. *Optica* **10**, 1195-1202 (2023). <https://doi.org:10.1364/OPTICA.498628>